

# 1 Impact of Internal Tides on Chlorophyll-a Distribution and Primary

# 2 Production off the Amazon Shelf from Glider Measurements and

# 3 Satellite Observations

Amine M'hamdi[1,2,8] , Ariane Koch-Larrouy[1,8] , Alex Costa da Silva[2] , Isabelle Dadou[1] , Carina Regina de
Macedo[1,3,7], Anthony Bosse[12] , Vincent Vantrepotte[3,2] , Habib Micaël Aguedjou[1,11] , Trung-Kien Tran[3] ,
Pierre Testor[9] , Laurent Mortier[10] , Arnaud Bertrand[4] , Pedro Augusto Mendes de Castro Melo[2] , James
Lee[5] , Marcelo Rollnic[5] , Moacyr Araujo[2,6]
[1]LEGOS, Université de Toulouse, CNRS, OMP, IRD, Toulouse, France.
[2]Departamento de Oceanografia, Universidade Federal de Pernambuco (DOCEAN/UFPE), Recife, Brazil.
[3]Univ. Littoral Côte d'Opale, CNRS, Univ. Lille, IRD, UMR 8187 - LOG - Laboratoire d'Océanologie et de Géosciences, F-
62930 Wimereux, France.
[4]MARBEC, Université de Montpellier, CNRS, Ifremer, IRD, Sète, France.
[5]Departamento de Oceanografia, Universidade Federal do Pará (UFPA), Belém, Brazil.
[6]Brazilian Research Network on Global Climate Change (Rede CLIMA), 12227-010, São José dos Campos-SP, Brazil.
[7]Earth Observation and Geoinformatics Division, National Institute for Space Research (INPE), São José dos Campos,
Brazil.
[8]CECI CNRS/Cerfacs/IRD, Université de Toulouse, Toulouse, France.
[9]LOCEAN-IPSL/CNRS, Université Pierre et Marie Curie, T45-55 E4 case 100, 4 place Jussieu 75252 Paris, France.
[10]Ecole Nationale Supérieure de Techniques Avancées, 29 rue d'Ulm,F-75230 Paris cedex 05, France.
[11]Centre National d'Etudes Spatiales, 18 av. Edouard Belin 31400 Tououse, France.
[12]Mediterranean Institute of Oceanography, OSU Institut Pytheas, Aix Marseille University, Université de Toulon, CNRS,
IRD, Marseille, France.
*Correspondence to*: Amine M'hamdi (abn.mhamdi@gmail.com)

## 24 Abstract.

The ocean region off the Amazon shelf including shelf-break presents a hotspot for Internal Tides (ITs) generation, yet its
impact on phytoplankton distribution remains poorly understood. These baroclinic waves, generated by tidal interactions with
topography, could modulate nutrient availability and primary production both by mixing and advection. While previous studies
have extensively examined the physical characteristics and dynamics of ITs, their biological implications—particularly in
nutrient-limited environments—remain underexplored. To address this question, we analysed a 26-day glider mission deployed
in September–October 2021 sampling hydrographic and optical properties (chlorophyll-a) at high resolution along an IT
pathway, satellite chlorophyll-a and altimetry data to assess mesoscale interactions. Chlorophyll-a dynamics were analysed
under varying IT intensities, comparing strong (HT) and weak (LT) internal tide conditions. Results reveal that ITs drive
vertical displacements of the Deep chlorophyll Maximum (DCM) from 15 to 45 meters, accompanied by a remarkable 50%
expansion in its thickness during HT events. This expansion is observed with a dilution of the chlorophyll-a maximum



concentration within the DCM depth. Turbulent cross-isopycnal exchanges driven by tides redistribute chlorophyll-a into adjacent layers above and below the DCM. At the surface, turbulent fluxes contribute to 38% of the chlorophyll-a supply, which directly influences primary production. Notably, the total chlorophyll-a content in the water column increases by 14-29% during high internal tide phases, reflecting a net enhancement of primary productivity. This increase results from the combined effect of vertical mixing and stimulated biological activity in the surface layer. These findings highlight the role of ITs as a key driver of chlorophyll-a distribution and short-term biological variability, reshaping the vertical chlorophyll-a profile and regulating primary productivity and potentially carbon cycling in oligotrophic oceanic systems.

## 1 Introduction

Internal Tides (ITs), also known as baroclinic tides, are ubiquitous in stratified oceans. These waves cause vertical displacements of isopycnal layers on the order of tens of meters and can propagate over long distances, reaching up to thousands of kilometers along the thermocline for the lowest modes (Zhao et al., 2016). Baroclinic tides are generated through the interaction of barotropic tidal currents with prominent submarine topographies such as continental slopes and mid-ocean ridges (Baines, 1982; Egbert and Ray, 2001; Munk and Wunsch, 1998). ISWs are highly stable internal waveforms that can propagate over long distances with a crest of a few tens of kilometers, and are generally structured with a wave train trailing behind the main crest (Alford et al., 2015; Jackson et al., 2012; Jeans and Sherwin, 2001). In addition to ITs, shorter-wavelength internal solitary waves (ISWs) may form from the nonlinear properties of ITs and dispersive processes, accompanying the ITs (Grimshaw, 2003; Grisouard et al., 2011). During their propagation, ITs and ISWs may eventually break down, releasing energy and driving vertical turbulent mixing (Alford et al., 2015; Lamb and Xiao, 2014; Moum et al., 2003; Nash et al., 2004). This mixing can play a crucial role in general circulation, contributing to the enclosure of the Atlantic Meridional Overturning Circulation (AMOC), and influencing oceanic energy and heat fluxes (Kantha and Tierney, 1997; Kunze, 2017; Waterhouse et al., 2014). Furthermore this mixing occurs close to the surface; it may also influence climate variability (Koch-Larrouy et al., 2010; Sprintall et al., 2014).

While the physical characteristics of ITs have been extensively studied, their impact on biogeochemical processes remains relatively poorly explored (Holligan et al., 1985; Liu et al., 2006; Ma et al., 2023; Zaron et al., 2023). Their influence on plankton dynamics is of significant interest, as phytoplankton constitutes the lowest trophic level of marine ecosystems. Through photosynthesis and organic carbon production, phytoplankton regulates primary productivity and influences global biogeochemical cycles (Falkowski and Knoll, 2007). The spatial and temporal variability of phytoplankton populations is driven by a combination of biological factors, such as production and grazing, and physical processes, including ocean currents, mesoscale structures (fronts and eddies), and heat fluxes (Mahadevan and Campbell, 2002; Van Gennip et al., 2016). Given the timescale and amplitude of disturbances generated by ITs, it is reasonable to hypothesize that ITs can significantly influence phytoplankton distribution.



The effects of ITs on phytoplankton could occur through, at least, two primary mechanisms. First, vertical mixing induced by
ITs can enhance nutrient fluxes into the euphotic zone, stimulating primary production and increasing phytoplankton biomass
in regions with high IT activity (Bourgault et al., 2011; Capuano et al., 2025; Horne et al., 1996; Kaneko et al., 2025; Law et
al., 2003; Lewis et al., 1986; Martin et al., 2010; Tsutsumi et al., 2020; Tuerena et al., 2019; Zaron et al., 2023). Second, the
vertical displacements associated with ITs can alter the light and nutrient conditions experienced by phytoplankton cells near
the pycnocline, thereby influencing their physiological responses and community structure (Gaxiola-Castro et al., 2002;
Holloway and Denman, 1989; Jacobsen et al., 2023; Kahru, 1983; Lande and Yentsch, 1988; Sangrà et al., 2002; Vázquez et
al., 2009)
The Amazon shelf-break is recognized as a hotspot for internal tide (IT) generation, dissipation and interact with intense
mesoscale features. First identified by Baines, 1982, subsequent studies have confirmed its role in converting barotropic energy
into baroclinic waves (Assene et al., 2024; Brandt et al., 2002; De Macedo et al., 2023; Ivanov et al., 1990; Magalhaes et al.,
2016; Tchilibou et al., 2022; Vlasenko et al., 2005). However, the specific impacts of these ITs on biological processes off
Amazon, particularly phytoplankton dynamics in the region, remain poorly understood and require further investigation.
This region, situated in the western tropical Atlantic near the mouth of the Amazon and Pará rivers, features a shallow
continental shelf and a macrotidal regime predominantly influenced by the semi-diurnal M2 tidal component(Beardsley et al.,
1995; Gabioux et al., 2005). The Amazon River significantly shapes local oceanographic conditions by modifying salinity,
temperature, and water column stratification (Geyer, 1995; Ruault et al., 2020). During the August-September-October (ASO)
season, reduced river discharge leads to a weaker and deeper pycnocline, along with a stronger North Brazil Current (NBC)
and higher eddy kinetic energy (EKE) (Neto and Da Silva, 2014; Silva et al., 2005; Tchilibou et al., 2022). The isopycnal
layers are thicker nearshore and become tighter offshore, causing weaker coastal stratification that increases offshore. These
seasonal variations clearly highlight the dynamic shifts in vertical density gradients, consistent with observations by Aguedjou
et al., 2019.
The dynamics of this region are further shaped by interactions with the NBC, a major western boundary current transporting
warm, saline waters from the South Atlantic (Garzoli et al., 2003; Johns et al., 1998; Schott et al., 1998; Silva et al., 2005).
Between June and February, the NBC undergoes a seasonal retroflection, forming large anticyclonic rings that propagate
northwestward (Fratantoni and Richardson, 2006,Fratantoni and Glickson, 2002).These anticyclonic eddies, known as "NBC
rings", can modulate stratification and nutrient distributions, influencing phytoplankton productivity (Mikaelyan et al., 2020).
During the second part of the year a large part of the NBC retroflects to feeds the eastward North Equatorial Countercurrent
(NECC) (Dimoune et al., 2023).
To investigate the role of ITs in-shaping phytoplankton dynamics in the oceanic region off the Amazon shelf, the AMAZOMIX
cruise aimed to collect a wide range of in situ measurements. Conducted between September and October 2021—an optimal



period for IT activity and mesoscale interactions—the cruise employs a multi-faceted approach combining numerical models,
satellite data, and in situ observations. In addition to ship-based measurements, an autonomous underwater glider was deployed
from September 9 to October 5, 2021 to have high resolution vertical structure data (hydrographic and chlorophyll-a)
influenced by ITs.
The objective of this study is to investigate how ITs influence the vertical distribution of Chlorophyll-a concentration off the
Amazon shelf. Analyses were performed by examining glider measurements and remote sensing observations, and by
comparing periods of strong and weak internal tide activity under similar stratification conditions.
**2 Data and Methods**
**2.1 Data**
**2.1.1 Autonomous glider**
On September 9, 2021, during the AMAZOMIX campaign an autonomous underwater glider (Testor et al., 2019)
was deployed for 26 days (09/09-05/10/ 2021) between the NBC and NECC, the adjacent oceanic waters off the
Northern Brazil (Figure 1) in the core of an ITs propagation path identified by Magalhaes et al., 2016 and Tchilibou
et al., 2022 . A Slocum G2 glider from Teledyne Webb Research was used, which is able to dive to 1000 m within
four hours and to cover approximately 20 km horizontally per day relative to the water. Due to strong currents near
1 m/s representing a real challenge for glider operation,  the glider only completes a total distance of 315 km over
ground during the 26-day deployment. The glider was equipped with a Seabird's pumped CTD (temperature,
pressure, conductivity), an Aanderaa optode (dissolved oxygen), and a WetLabs's optical puck (chlorophyll-a
fluorescence, CDOM, and turbidity). The sensors had a sampling frequency of 5 seconds, resulting in a vertical
sampling interval of approximately 1 m. Between each surfacing, the glider estimates its position thanks to
navigation sensors (compass) enabling to estimate a mean dive-average horizontal currents while comparing its
dead-reckoned position with GPS fixes. The glider dataset was processed using the Geomar Matlab Toolbox
(Krahmann, 2023), which includes the removal of thermal lag errors following (Garau et al., 2011). Temperature
and salinity were converted to conservative temperature and absolute salinity using the Gibbs Seawater python
library (McDougall and Barker, 2011). The temperature and salinity profiles were validated by comparison with a
reference CTD at the glider deployment site. Daytime chlorophyll-a fluorescence profiles were corrected for non-
photochemical quenching processes using the method described by (Thomalla et al., 2018), setting the quenching
depth at 40 m. To enable direct comparison with satellite-derived data, chlorophyll-a concentrations measured by



the glider were averaged from the surface down to the first optical depth (Zpd = Zeu/4.6 Morel, 1988) to build the
time series, which defines the depth range primarily sensed by ocean colour remote sensors.

**Figure 1 : Chlorophyll map averaged between 09/09/21 and 05/10/21 in the glider deployment region, divided into four subregions: A (blue), B (red), C (green), and D (magenta), each characterized by distinct temperature–salinity (T/S) properties (section 3.1). The yellow area marks the main surface current, purple indicates the plume, and light brown highlights AE1, the anticyclonic eddy detected by altimetry during the transect. The white dashed line shows the main internal tide propagation pathways identified by Magalhães et al. (2016) and Tchilibou et al. (2022), while the grey circle marks the primary internal tide generation site (46° / 0.75°).**



**2.1.2 Remote sensing observations: ISW detection, Chlorophyll-a distribution and Mesoscale eddy tracking**

Internal solitary waves (ISWs) create patterns alternating between rough and smooth surface areas, corresponding to convergent and divergent surface currents, respectively. Thus, their signatures in MODIS images during sunglint or in the SAR imagery are manifested by variations in sea surface roughness, resulting in changes in the brightness of the captured images (De Macedo et al., 2023; Jackson and Alpers, 2010; Magalhaes et al., 2016). During the cruise, ISW signatures were visually identified and manually extracted off the Amazon shelf from a representative assembled data set composed of 21 remote sensing imagery acquired by active and passive sensors from 1st September 2021 to 10th October 2021. A total of 13 imagery were acquired by the synthetic aperture radar (SAR) C-band (centre frequency of 5.4 GHz) Copernicus Sentinel-1A and 1B instruments Level-1 GRD (ground range detected) products in the interferometric wide swath mode with about 250 km of swath and spatial resolution of 20.3 x 22.6 m (range x azimuth), operating in single polarization (VV channel). The SAR imagery were collected from the Copernicus Open Access Hub (https://scihub.copernicus.eu/dhus/#/home). The SAR scenes were pre-processed using the software SNAP and Sentinel Toolboxes (version 8.0) by calibrating the data (conversion from digital number to normalised radar cross-section) and applying a 5x5 boxcar filter to reduce the speckle noise. A total of 8 Level 1B imagery were acquired by the Moderate Resolution Imaging Spectroradiometer (MODIS) sensor onboard the TERRA and AQUA satellites. The band 6 centred at 1640 nm with a spatial resolution of 500 m was used to identify the ISW signatures in the sun glint region. The MODIS/TERRA and AQUA imagery were collected from NASA's Earth Science Data System, ESDS (https://earthdata.nasa.gov/).

Given that ocean colour observations are often affected by interference from clouds, leading to data gaps, we used the daily mean merged chlorophyll-a (with a spatial resolution of ~4 km) product from the GlobColour project to maximise data coverage from 1st September 2021 to 05 th October 2021 provided by the ACRI-ST company (Garnesson et al., 2019). This product provides chlorophyll-a concentration information from the ocean colour sensors MODIS-AQUA, NPP-VIIRS, NOAA20-VIIRS, and Sentinel-3 OLCI A and B, including updated ancillary information (i. e., meteorological and ozone data for atmospheric correction, and attitude and ephemerides for data geolocation). According to Garnesson et al., 2019, the approach merge three algorithms: 1) the CI approach for oligotrophic waters (Hu et al., 2012) ; 2) the OCx (OC3, OC4 or OC4Me depending on the sensor) for mesotrophic waters; and 3) the OC5 algorithm for complex waters (Gohin, 2011). The product can be found on the CMEMS website (https://resources.marine.copernicus.eu/products).In this study, we utilized data provided by GlobColour, specifically estimates of the euphotic depth (Zeu) derived from satellite observations of the MODIS-Aqua sensor. These Level-3 processed data are available at a spatial resolution of 4 km and were obtained from the GlobColour platform. The euphotic depth was estimated following the methodology described by Morel and Maritorena (2001), which defines Zeu as the depth where incident light is reduced to 1% of its surface value. The dataset is publicly available at HERMES ACRI.





Daily maps of the Ssalto/Duacs absolute dynamic topography (ADT) gridded product were used to identify and track coherent
mesoscale eddies during AMAZOMIX campaign. This product was obtained from all available satellite altimetry along-track
data and optimally interpolated onto a 0.25°×0.25° longitude/latitude (Pujol et al., 2016).The product can be found on :
(https://data.marine.copernicus.eu/product/SEALEVEL_GLO_PHY_MDT_008_063/description )
Mesoscale eddies were identified, using the algorithm developed by Chaigneau et al., 2009, 2008; Pegliasco et al., 2015. In
this method, an eddy is identified by its centre and its external edge. An eddy centre corresponds to a local extremum
(maximum for an anticyclonic eddy and minimum for a cyclonic eddy) in ADT while eddy edge corresponds to the outermost
closed ADT contour around each detected eddy centre. One long-lived anticyclonic eddy (AE1) was identified during the
AMAZOMIX campaign. AE1 was generated within the study domain from instability of NECC and propagated north
westward making NECC oscillating the NBC figure 3 . AE1 lasted more than 120 days. The bathymetric data used in this
study are sourced from the NOAA CoastWatch Program and are accessible through the NOAA CoastWatch Data Portal. These
data are formatted for MATLAB and are stored under the directory gov.noaa.pfel.coastwatch.Matlab. The bathymetric dataset,
referenced from the Topography SRTM30 Version 6.0 (30 Arc-Second Global), provides detailed seafloor topography
information crucial for analysing oceanographic processes. Additionally, the geostrophic velocity data used in this study are
sourced from the Global Ocean Gridded SSALTO/DUACS Sea Surface Height L4 product, provided by Mercator through the
Copernicus Marine Service. This product includes surface geostrophic eastward and northward sea water velocities, calculated
from sea surface height assuming sea level as the geoid reference. These data, derived from sea surface height, provide essential
surface currents.The dataset is available via Copernicus Marine Data.

### 2.1.3 FES model

Tidal data were extracted from the global FES2014 (Finite Element Solution) model developed by Lyard et al., 2021. The
outputs of the sea surface elevation field (eta) were used at the grid point corresponding to 46°N, 0.75°E, which corresponds
to an internal tide generation site previously identified by Magalhaes et al., 2016 and Tchilibou et al., 2022. The use of those
data helped us to identify neap tides and spring tides.

### 2.2 Methods

To assess the impact of ITs (ITs) on the vertical distribution of chlorophyll-a (hereafter referred as CHL for the equations), a
multi-step approach was applied. (1) Satellite observations were used to characterize the large-scale spatial distribution of
chlorophyll-a and the physical processes influencing it, enabling the identification of hydrographically distinct regions (section
3.1) (2) Based on this preliminary analysis, glider data were divided into transects corresponding to periods with contrasting
hydrographic properties named A, B, C and D (Fig 1) (Section 3.2). (3) Given the prevalence of ITs in the study area, A and
B period was further subdivided into low tide (LT) and high tide (HT) phases using spectral analysis of the temperature field
to estimate tidal amplitude; the classification was based on the presence of a spectral peak at the M2 frequency (section 3.2).
(4) Chlorophyll-a fluorescence profiles collected by the glider were then averaged and statistically compared between LT and



HT conditions to evaluate the effect of IT intensity on chlorophyll-a vertical distribution (section 3.3). (5) Finally, vertical
turbulent fluxes of chlorophyll-a were estimated to better understand the transport mechanisms associated with ITs (section
3.3 ).

### 2.2.1 Temperature Power Spectra

We analysed temperature time series between 145m and 165 m depth, where the largest vertical displacement of isotherms
was observed. The high-frequency glider profiling (about 12 profiles per day) enabled the construction of temperature time
series resolving the main tidal frequency (12h). All temperature measurements between 145 and 165 meters were aggregated
into a single time series. The aggregated time series was resampled at 30 mins to ensure regularly spaced data points and
detrended to remove long-term variations. A Fast Fourier Transform (FFT) was then applied to convert time series into
frequency domain. The power spectrum was calculated to identify the dominant frequencies of oscillations  (McInerney et al.,
2019).

### 2.2.2 Diapycnal chlorophyll fluxes estimation

The  vertical dynamics of chlorophyll-a concentration (CHL) in the water column is described by the following
equation :

$$\frac{\partial CHL(z,t)}{\partial t} + w\frac{\partial}{\partial z}CHL(z,t) = \frac{\partial}{\partial z}\left(Kz\frac{\partial}{\partial z}CHL(z,t)\right) + SMS(z,t) \quad (1)$$

Where CHL is the  chlorophyll-a concentration. The $w\frac{\partial}{\partial z}CHL(z,t)$ term represents the vertical advection of chlorophyll by
the vertical velocity field w, while $\frac{\partial}{\partial z}\left(Kz\frac{\partial}{\partial z}CHL(z)\right)$ accounts for vertical turbulent diffusion, with $K_Z$ being the diffusivity
coefficient. The  source-minus-sink (SMS) term encompasses biological processes, specifically primary  production and
grazing, which regulate the net chlorophyll-a balance in the system.
To isolate turbulent chlorophyll-a fluxes, the analysis is conducted within a vertical isopycnal reference framework. In this
context, the advection term $w\frac{\partial}{\partial z}CHL(z) = 0$ as vertical velocities advect isopycnals up and down. By changing the vertical
coordinate from z to rho $\frac{\partial CHL(\rho(z))}{\partial z} = \frac{\partial \rho}{\partial z}\frac{\partial CHL(\rho(z))}{\partial \rho}$ and assuming $\frac{\partial \rho}{\partial z}$ and $K_z$ is constant leads to  the equation:

$$\frac{\partial CHL(\rho,t)}{\partial t} = K_v \frac{\partial^2 CHL(\rho,t)}{\partial \rho^2} + SMS(\rho) \quad (2)$$



Where the constant $K_v = (\frac{\partial \rho}{\partial z})^2 K_z = (\frac{N^2 \rho_0}{g})^2 K_z$ represents the diapycnal diffusivity coefficient with $\rho_0$ the mean density of
the ocean and g the gravitational acceleration . By integrating between two isopycnal density surfaces ($\rho_0$ and $\rho_1$), the average
variations over a given period ($\Delta T$) are defined as:

$$< P>_{\rho 0, \rho 1, \Delta T} = \frac{1}{\Delta T} \int_{\Delta T} \int_{\rho 0}^{\rho 1} \frac{\partial P(\rho, t)}{\partial t} d\rho dt \quad (3)$$


Where P correspond  either to $\frac{\partial CHL(\rho, t)}{\partial t}$ ; $K_v \frac{\partial^2 CHL(\rho, t)}{\partial \rho^2}$ ; $SMS(\rho)$ and $<P>$ to $<CHL>$, $<DIFF>$ or $<SMS>$
For two distinct periods corresponding to complete tidal cycles with intense tides $\Delta HT$ and low tides $\Delta LT$, and within
a density layer between $\rho_a$ and $\rho_b$, the differences are defined as :

$$\Delta P_{\rho 0, \rho 1, Tides} = < P >_{\rho 0, \rho 1, HT} - < P >_{\rho 0, \rho 1, LT} \quad (4)$$


The comparison between periods of strong (HT) and weak (LT) tidal forcing, relating to spring tides / neap tides cycle,
conducted in a region with similar hydrodynamic properties but primarily differentiated by the intensity of ITs (ITs), served
as a proxy for quantifying the influence of ITs on turbulent chlorophyll-a fluxes.
We divided the water column into three isopycnal layers: the surface layer, the Deep Chlorophyll Maximum (DCM layer), and
the bottom layer. We assumed that the difference of mean CHLa integrated in DCM at the DCM ($\Delta Diff_{DCM}$) is redistributed
upward and downward through mixing, with proportions n for the surface layer and m for the bottom layer, where n+m=1.
Using this partitioning approach, we express the variation in chlorophyll-a ($\Delta CHL$) for each layer as follows:

$$\Delta CHL_{SURF} = -n . \Delta Diff_{DCM} + \Delta SMS_{SURF} \quad (5)$$


$$\Delta CHL_{DCM} = \Delta Diff_{DCM} + \Delta SMS_{DCM} \quad (6)$$


$$\Delta CHL_{DEEP} = -m . \Delta Diff_{DCM} + \Delta SMS_{DEEP} \quad (7)$$


With $-n . \Delta Diff_{DCM} = \Delta Diff_{SURF}$ and $-m . \Delta Diff_{DCM} = \Delta Diff_{DEEP}$
By summing Equations 5, 6, and 7, the diffusion-related component cancels out, leaving:

$$\Delta SMS_{DCM} + \Delta SMS_{Surf} + \Delta SMS_{Deep} = \Delta CHL_{TOT} \quad (8)$$




The total chlorophyll-a variation $\Delta CHL_{TOT}$ between high tide (HT) and low tide (LT) periods is interpreted as follows:
if $\Delta CHL_{TOT} > 0$ this value represents the minimum possible net production. Respectively if $\Delta CHL_{TOT} < 0$ it indicates a
dominance of grazing

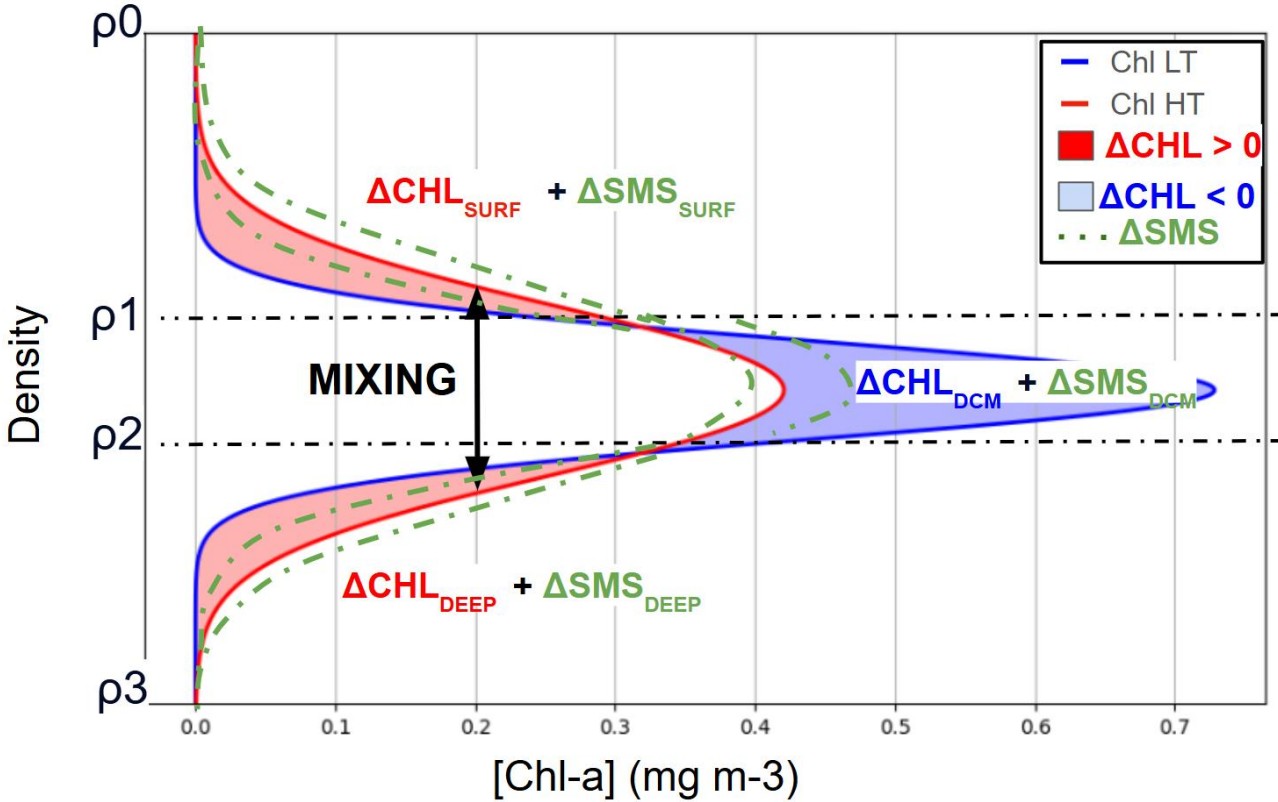


**Figure 2 : Schematic of Vertical diffusion of chlorophyll-a Peak between LT period (blue) vs HT period (red) with profile**
**modification due to ITs mixing**
**2.2.3 Statistical Analysis**
In this study, various statistical methods were employed to analyse the impact of ITs on chlorophyll-a distribution across
density layers. The Mann-Whitney U test, a non-parametric test, was selected to compare chlorophyll-a concentrations between
periods of high and low ITs within different density layers. This test is particularly suitable here, as it does not require the
assumption of data normality distribution, which is often difficult to ensure for environmental samples with irregular
distributions. Mean comparisons and percentage changes provide a statistical approach of ITs on chlorophyll-a. Maximum
chlorophyll-a concentrations and DCM thickness were extracted from fluorescence profiles. A Pearson correlation analysis
was performed to assess the linear relationship between these variables. Statistical significance was determined using the



associated p-value.Additionally, descriptive statistics by isopycnal layer were calculated for each density zone, offering a
detailed view of trends specific to layers and enabling the identification of significant changes. Collectively, these methods
robustly capture significant differences and their potential effects on chlorophyll-a distribution and concentration.

## 3 Results

### 3.1 The glider study area

The oceanic circulation in the study area was dominated by two major current systems: the NBC and the NECC.Their
interaction was regulated by the seasonal retroflection of the NBC, as that was clearly illustrated in the Absolute Dynamic
Topography (ADT) maps (Fig. 3a–d). This circulation was associated with ADT values reaching approximately 0.6 m. From
a biogeochemical perspective, strong contrasts were observed between offshore waters and the Amazon continental shelf.
The offshore waters were characterized by oligotrophic conditions, with low chlorophyll-a concentrations (~0.1 mg m-3),
whereas the Amazon shelf was dominated by turbid waters, rich in suspended matter, with chlorophyll-a concentrations
exceeding 1 mg m-3. This gradient highlighted the significant influence of the Amazon plume on local productivity.
Moreover, the depth of the euphotic layer (Zeu) (Fig.4 purple) remained relatively stable along the glider transect, ranging



between 72 m and 87 m.





**Figure 3. (a–d) Absolute Dynamic Topography (ADT) maps for September 11 (a), 16 (b), 22 (c), and 28 (d), 2021. (e–h) Satellite-derived surface chlorophyll-a maps for the same dates: September 11 (e), 16 (f), 22 (g), and 28 (h). The AE1 eddy is outlined by white ovals. The glider trajectory is shown as a grey line, with color-coded segments indicating periods A, B, C, and D (as defined in Fig. 1). Geostrophic surface currents are shown as arrows. The 1000 m isobath is marked in green (a–d) and red (e–h).**

*Formation and Evolution of the Anticyclonic Eddy (AE1)*

On Sept 11st 2021, an anticyclonic eddy (AE1) formed in the region, identified by an ADT peak reaching approximately 0.7 m (Fig. 3a, white circle). The eddy core gradually migrated from 44.5°W -4°N to 47.5°W-5.5°N over the following 27 days,





covering roughly 372 km with an average speed of 0.16 m/s. Between Sept 12nd and Sept 19th, the eddy underwent significant
expansion, with its radius increasing from 100 km to approximately 400 km.

*Glider-Eddy Interactions*

The influence of the eddy on surface velocity is evident from an initial decrease in speed from 0.58 m/s to 0.17 m/s in Sept
17th, followed by a gradual acceleration reaching 0.8 m/s at the end of the transect (Fig. 4, bottom panel). Between Sept 14th
and Sept 22nd, as the glider traversed the eddy, variations in its distance from the eddy's outer boundary were observed (Fig.
4, top panel). These fluctuations confirm that the glider remained along the eddy's periphery, highlighting the kinematic effects
induced by its circulation. Maximum geostrophic velocities, derived from ADT gradients, further indicate intense eddy
dynamics, with circulating currents reaching up to 0.8 m/s toward the end of the observation period.

*Biogeochemical Characteristics Associated with AE1*

The lowest Chlorophyll values (~0.11 mg m-3) along the glider acquision were recorded in the eddy core, which was
characterized by minimal velocities and maximum Absolute Dynamic Topography (ADT). In contrast, higher biological
activity was observed at the eddy's periphery, marked by dashed black lines on September 14 and Sept 22$^{nd}$ (Fig. 4),
emphasizing the spatial heterogeneity induced by the eddy's circulation. This pattern was explained by the typical behaviour
of anticyclonic eddies, where isopycnal depression inhibited the upward flux of nutrients, thereby limiting primary
productivity. The coupling between the eddy's physical dynamics and the distribution of biological parameters was highlighted
by the chlorophyll-a maps. During the eddy-impacted period (shaded in grey), both glider (dashed green) and satellite (solid
green) chlorophyll-a data show a decrease at the eddy center (Sept 16th – Sept 19th) and an increase at its edge (Sept 14$^{th}$ and
Sept 22$^{nd}$). In addition to the smoother satellite signal, the glider data reveal short-term oscillations; these high-frequency
variations, likely associated with diurnal and semidiurnal processes, are not resolved by satellite observations, although the
satellite successfully captures the overall trend and order of magnitude.

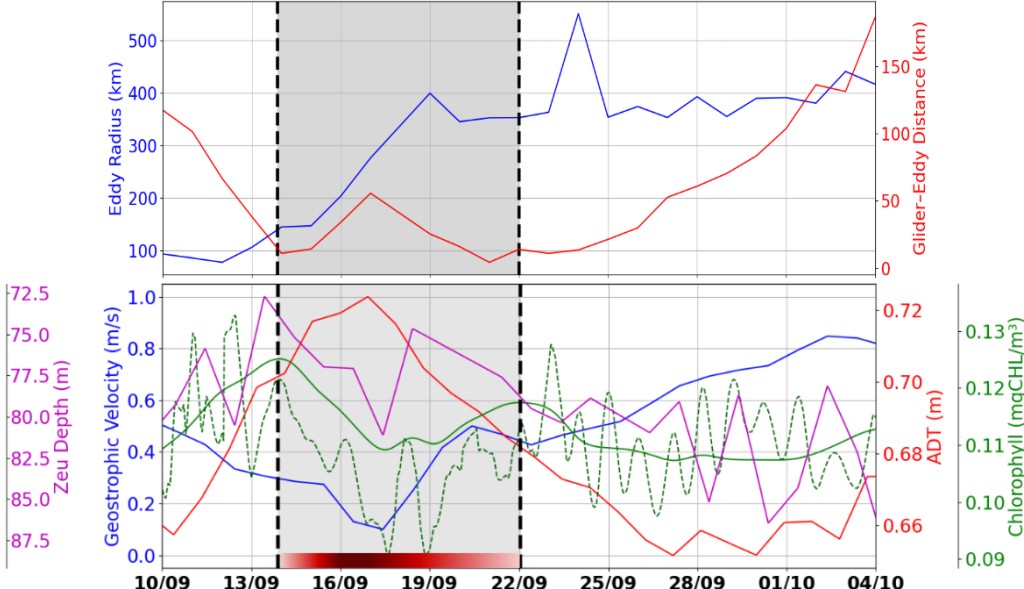

**Figure 4: (Top) Time series of the distance between the glider and the nearest eddy contour (red) and the maximum eddy radius (blue). (Bottom) Geostrophic velocity magnitude along the glider's track (blue), ADT along the glider's track (red), chlorophyll-a concentration along the glider's track from GlobColour (solid green), integrated chlorophyll-a between surface and Zpd from glider (dashed green), euphotic depth along the glider's track (purple). The red segment represents AE1, with shading that becomes lighter towards the edge and darker at the core**

*Internal Solitary Waves*

During the observation period, between Sept 9th and Sept 23rd 2021, a total of 12 internal solitary wave (ISW) crests were identified (Table 1). These waves were detected through a combination of satellite observation and in situ glider measurements, enabling documentation of their occurrence and dynamics over a two-week period. Satellite images (Fig. 5a–c), acquired on Sept 9th and Sept 11th (sunglint information) and on Sept 23rd (SAR imagery), revealed the surface signatures of internal solitary waves. The glider's position, marked by an orange cross on the images, confirms the influence of intense ISWs during its evolution. Figure 5d illustrates the tidal current amplitudes derived from the FES2014 model (Lyard et al., 2014) at the point (46°W-0.5°N). The graph highlights the variations between spring tides (blue-shaded areas) and neap tides (white areas), as well as transitional phases (yellow). The black rectangles in Fig. 5d, indicating the occurrences of solitons, show a clear alignment between the presence of internal wave trains and spring tide periods, as also shown by De Macedo et al. (2023). The observed waves primarily propagated towards the northeast, with wavelengths ranging from 117 km to 175 km, characteristic of mode-1 internal waves. These structures exhibit rapid dynamics, with estimated propagation speeds of approximately 3 m/s (De Macedo et al., 2023), significantly faster than the average speed of the glider (~0.14 m/s). This speed difference justifies

that the glider is unable to capture the same wave crest more than once. and is almost stationary in the ITs field (reduced

aliasing).

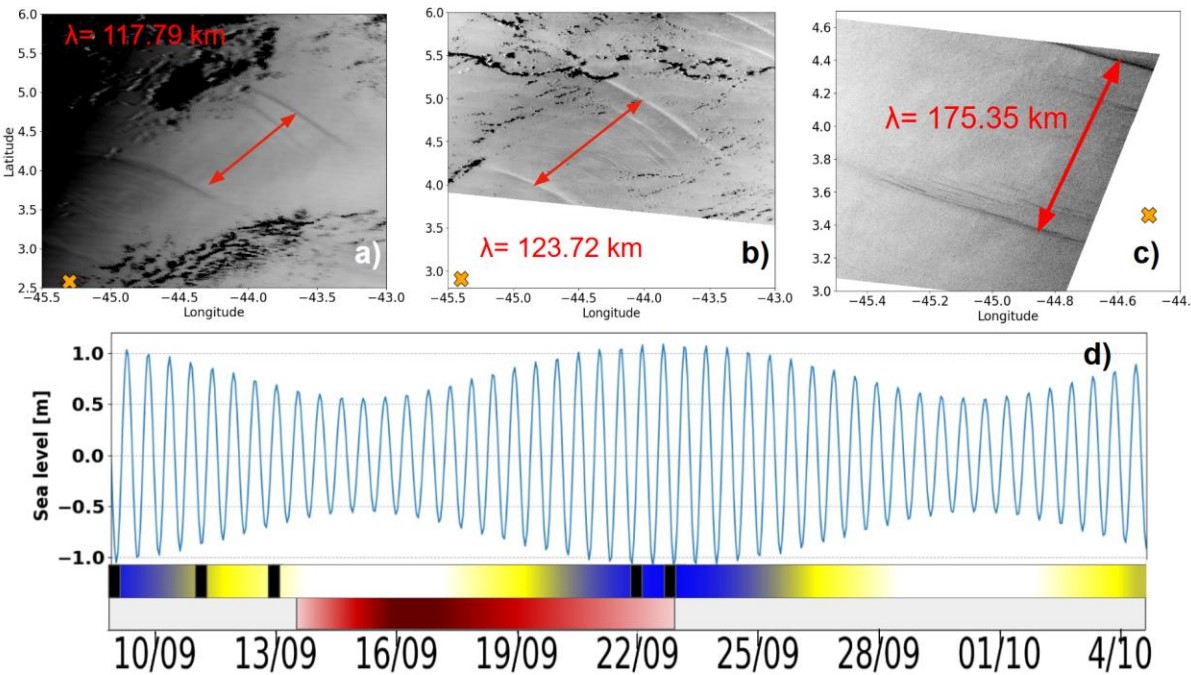

**Figure 5: (a-c) Satellite imagery acquired on September 9, 2021, at 13:45 UTC, and September 11, 2021, at 13:30 UTC (both from**

**sunglint imagery), as well as on September 23, 2021, at 08:47:35 UTC (SAR imagery). (d) Tidal current amplitudes derived from**

**the FES2014 model (Lyard et al., 2021) at point (46°W,0.5°N). The orange cross denotes the glider's position at the corresponding**

**timestamps. The timeline illustrates the variation of Spring Tides (blue), Neap Tides (White), transient zone (Yellow). The red**

**segment represents AE1, with shading that becomes lighter towards the edge and darker at the core.**

**Table 1: Internal solitary waves detected by SAR (Sentinel-1) andor Sunglint (Modis) during the period Sept 9[th] to Oct 5[th] off Amazon**

**self/shelf-break region**





| Date | Sensor Type | Crest |
|------|-------------|-------|
| Sept 9th 2021 | MODIS | 1 |
| Sept 11st 2021 | Sentinel-1 | 3 |
| | MODIS | 1 |
| Sept 13rd 2021 | MODIS | 4 |
| Sept 22nd 2021 | MODIS | 1 |
| Sept 23nd 2021 | Sentinel-1 | 2 |

**3.2 Physical Near Surface Processes**

*Near Surface Hydrography*

The hydrographic observations collected by the glider between the surface and 200 m depth reveal a strongly stratified thermohaline structure, characteristic of tropical waters (Fig. 6). The temperature (T), salinity (S), and density ($\sigma_0$) profiles indicated the presence of a homogeneous layer between 0 and 50 m, followed by a thermocline, halocline, and pycnocline extending from 50 to 160 m. Salinity above 35.5 in this region indicates euhaline conditions, showing the plume did not affect the southern area.. Notable hydrographic changes are further observed during the study period. Between Sept 14th and Sept 22nd, as the glider crossed AE1 (marked by circular arcs in Fig. 3a), a lenticular and homogeneous isopycnal field was detected, with nearly uniform temperature (27°C) and salinity (36.5) between isopycnals 23.5 and 23.7 (75m–125m depth). At the surface, this anticyclone exhibited warmer and more stratified waters compared to the surrounding environment, while salinity remained homogeneous. Prior to crossing AE1, the glider was deployed at the edge of the NBC (Fig. 1, Fig. 3a) from September 9th to 14th. In contrast, this region displayed a homogeneous temperature layer but a stratified salinity profile. During this period, a maximum salinity zone (~36.7) was observed between 120 and 150 m depth, generally associated with the maximum salinity transport by the North Brazil Current (NBC) (Silva et al., 2009), which was significantly reduced in subsequent periods indicating the shift in background conditions. Between September 22nd and 28th, the region was characterized by a homogeneous surface layer (0–50 m) in both temperature and salinity, accompanied by a pronounced uplift of the 23.10 isopycnal. Below 50 m, the ocean became increasingly stratified, exhibiting coherent variations in T and S. From Sept 28th to



Oct 4th, the glider entered the waters of the North Equatorial Countercurrent (NECC), characterized by an accelerated geostrophic current field (Fig. 4, bottom), reaching speeds of 0.6 m/s eastward. The hydrography during this period revealed the warmest surface layer of the transect (~30℃), followed by a coherent stratification in T and S. The four distinct regions, identified through these hydrographic variations, have been designated as A, B, C, and D, while the edges have been excluded, as they are considered transition periods.

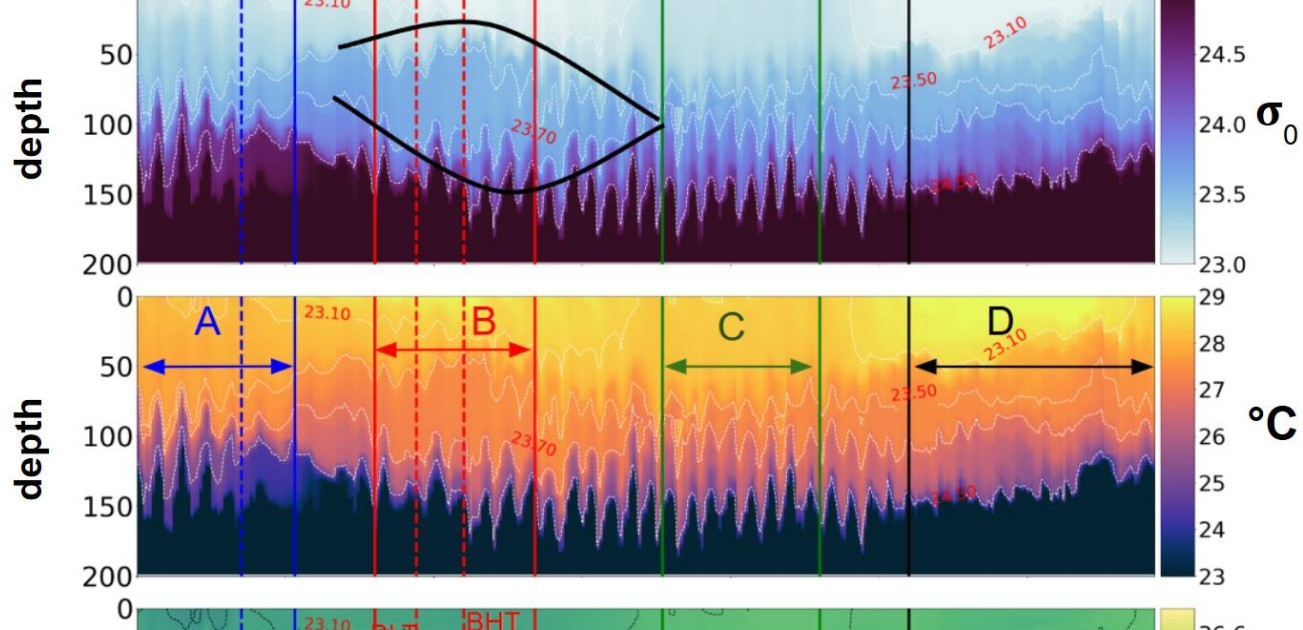



**Figure 6: Hovmöller diagrams showing (a) $\sigma_0$ density (with the black lens highlighting eddy AE1 as identified in Section 3.1), (b) Conservative Temperature, and (c) Absolute Salinity, all derived from glider observations. The timeline below the plots indicates key oceanographic processes discussed in Section 3.1: blue segments mark spring tide events, white indicates neap tides, and the red segment corresponds to AE1, with shading intensity increasing from lighter at the periphery to darker at the core. A black rectangle marks the occurrence of internal solitary waves (ISWs) detected from satellite data. Labels A, B, C, and D denote distinct periods, A and B further divided into high tide (HT) and low tide (LT) subperiods, reflecting variations in tidal intensity.**

*Transect Divided into Four Periods*

To assess the impact of ITs on chlorophyll-a, the transect was divided into distinct periods based on hydrographic criteria, ensuring a robust comparative framework. The relevance of this classification (A, B, C, and D) was validated using T/S diagrams (Fig. 7), where four distinct hydrographic profiles were identified. Period A (Sept 9th – Sept 13rd, Blue) was characterized by a strong salinity gradient above 23.52 kg/m³, ranging from 36.2 to 36.6, while temperature remained stable around 28°C. This period was observed at edge of the NBC (Fig. 1, Fig. 3), with a total distance of 96.69 km covered by the glider. Period B (Sept 15–19, Red) was located within (AE1), where a well-defined T/S stratification was observed, indicating a stable water mass structure. During this phase, a distance of 84.20 km was recorded. Period C (Sept 22nd –25th, Green) was identified as a transition zone between B and D, exhibiting a structure similar to Period B in the 23.3 –24 kg/m³ layer but with a saltier water mass in the 24–24.8 kg/m³ range. Period D (Sept 28th –Oct 5th, Black) was associated with waters in the influence of the North Equatorial Countercurrent (NECC) (Fig. 1, Fig. 3), where the T/S profile revealed three distinct layers within the 0–200 m column. The surface layer (23–23.3 kg/m³) was stratified in temperature while salinity remained constant (~36.3). Beneath it, an intermediate transition layer (23.3–23.7 kg/m³) was marked by coherent T and S gradients, followed by a deep layer, where temperature remained stratified, and salinity was stable (~36.5). This classification was found to effectively capture the hydrographic variability along the transect, providing a general frame for analysing internal tide dynamics.



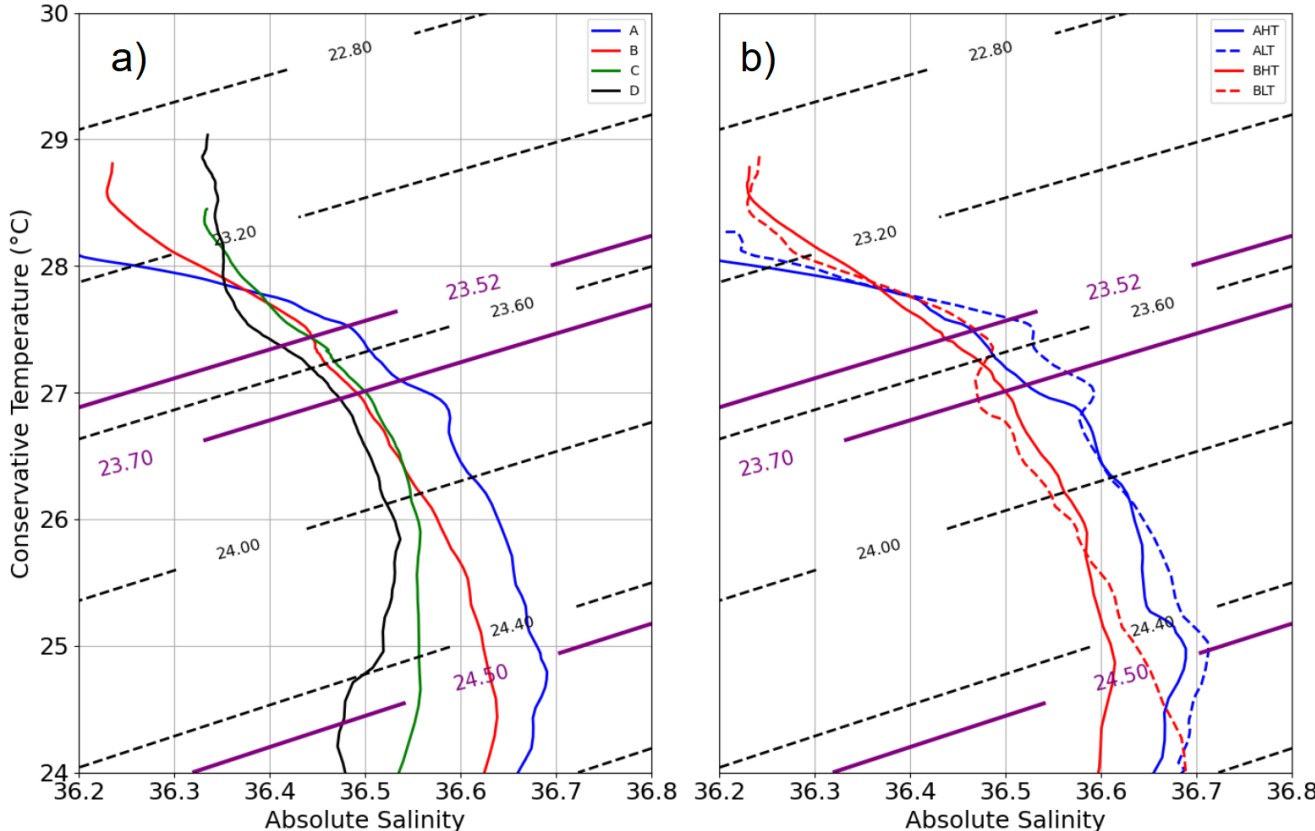

**Figure 7: T/S Diagram (a - left) for periods A, B, C, and D and (b - right) for the sub-periods High Tides (HT) / Low Tides (LT) within periods A and B. Black dotted isopycnals are plotted at intervals of 0.4 kg/m3.**

*Thermocline Oscillations Driven by ITs: Variability Between High and Low Tides*

Thermocline oscillations were observed in all periods except D, with amplitudes ranging from 10 to 50 m and peaking near the 24.5 isopycnal (Fig. 6). These in-phase oscillations were most intense at the pycnocline and gradually diminished toward the surface, and were modulated by neap and spring tide cycles, with peaks coinciding with Internal Solitary Wave (ISW) events (Fig. 5). Spring tides (A and C) induced a ~1°C temperature drop, contrasting with the surface warming in period D, when no oscillations were detected. A Fast Fourier Transform (FFT) analysis of isotherms (145–165 m) (McInerney et al., 2019)confirms the semi-diurnal modulation of these oscillations. Periods A and B were further divided into high Tide-amplitude (AHT, BHT) and low Tides-amplitude (ALT, BLT) phases, while Period C exhibits continuous oscillations. All showed a 12h25 spectral peak (Fig. 8) with a tenfold increase in spectral power, confirming the influence of ITs. Furthermore, Figure 7b reveals that these oscillatory phases correspond to the same water masses, validating the subdivision AHT/ALT and BHT/BLT.





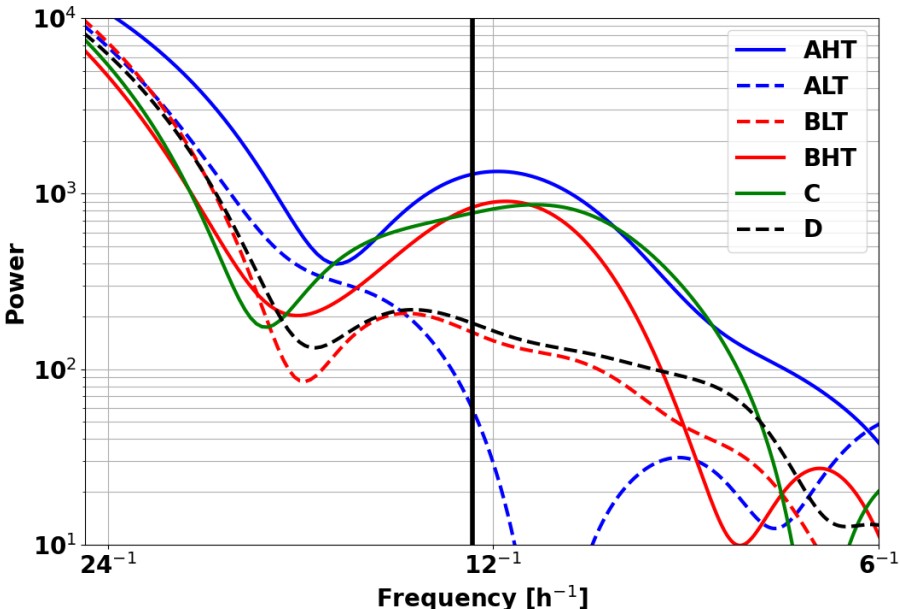

**Figure 8**: Spectral Analysis of temperature time series in Regions A, B, C, and D, across 145m and 165m depth.

**3.3 ITs effect on chlorophyll**

*Overview of Subsurface Processes Effects on Chlorophyll*

The vertical distribution of chlorophyll-a along the transect (Fig. 9a) was characterized by a three-layer structure. A Deep Chlorophyll Maximum (DCM) was observed between isopycnals 23.53 and 23.7 (corresponding to depths of approximately 70m and 120 m), with concentrations ranging from 0.4 to 0.8 mg m-3. The lowest value (0.4 mg m-3) was recorded during period B, coinciding with the passage of the glider through the anticyclonic eddy AE1, in agreement with the surface signal (Fig. 4, green). At the edges of AE1, a slight uplift of the DCM was observed, attributed to the upward displacement of isopycnals. Above 23.53, a surface layer was identified, while a deeper layer extends between 23.7 and 24.5. A key finding is the influence of ITs on the vertical structure of chlorophyll-a. The tides induced DCM oscillations with amplitudes between 15 and 45 meters at depths of 65 to 125 meters during AHT, BHT, and C, while weaker disturbances were observed during ALT and BLT (Fig. 9b). These disturbances could impact primary production, as the light gradient is non-linear — an uplift exposes the chlorophyll-a layer to more light than the amount lost by a downlift. The following section now focuses on the characterization and quantification of ITs processes advection and mixing, that influence chlorophyll-a distribution.






**Figure 9: (a)** Hovmöller diagram of chlorophyll-a distribution from 0 to 200 m between September 9, 2021, and October 4, 2021. Dark green segments indicate spring tide events, while light green segments correspond to neap tides. The red segment represents Anticyclonic Eddy 1 (AE1), with lighter shades at the edges and a darker core. The black rectangle highlights the presence of Internal Solitary Waves (ISWs). **(b)** Hovmöller diagram of chlorophyll-a from 0 to 200 m, segmented by tidal phases: ALT, AHT, BLT, and BHT.

*Variability of Chlorophyll-a Structure Between High and Low ITs*



Higher chlorophyll-a concentrations were found at the surface and in deeper layers during HT, while chlorophyll-a
concentration was more pronounced within the DCM during LT (Fig. 10). To assess these variations and evaluate the impact
of ITs on the vertical redistribution of chlorophyll, four key parameters were examined: maximum chlorophyll-a concentration,
chlorophyll-a peak thickness, Total averaged chlorophyll-a content, and DCM depth (Table 2). The chlorophyll-a peak
thickness corresponds to the depth range where concentrations exceed 0.2 mg m-3.
Under HT conditions, ITs induce vertical displacements of chlorophyll, leading to a redistribution of biomass across different
layers. Specifically, maximum chlorophyll-a concentration decreased by 17% (0.12 mg m-3) in period A (resp. 9%, 0.04 mg
m-3 in period B), while the peak thickness expanded by 50% (resp. 30%). The resulting redistribution led to an inverse
relationship between maximum chlorophyll-a concentration and DCM thickness (Fig. 11). This correlation was statistically
significant, with Pearson coefficients of r = -0.43 (p = 0.015) for period A and r = -0.31 (p = 0.026) for period B. Total averaged
chlorophyll-a content increased significantly during HT, with rises of 14% ($\Delta CHL_{total}$ = 4.44 mg m-2 where $\Delta CHL_{total}$ is the
total variation of averaged integrated chlorophyll-a between HT and LT) in period A (resp. 29%, $\Delta CHL_{total}$ = 6.98 mg m2 in
period B), indicating an overall enhancement of primary production.

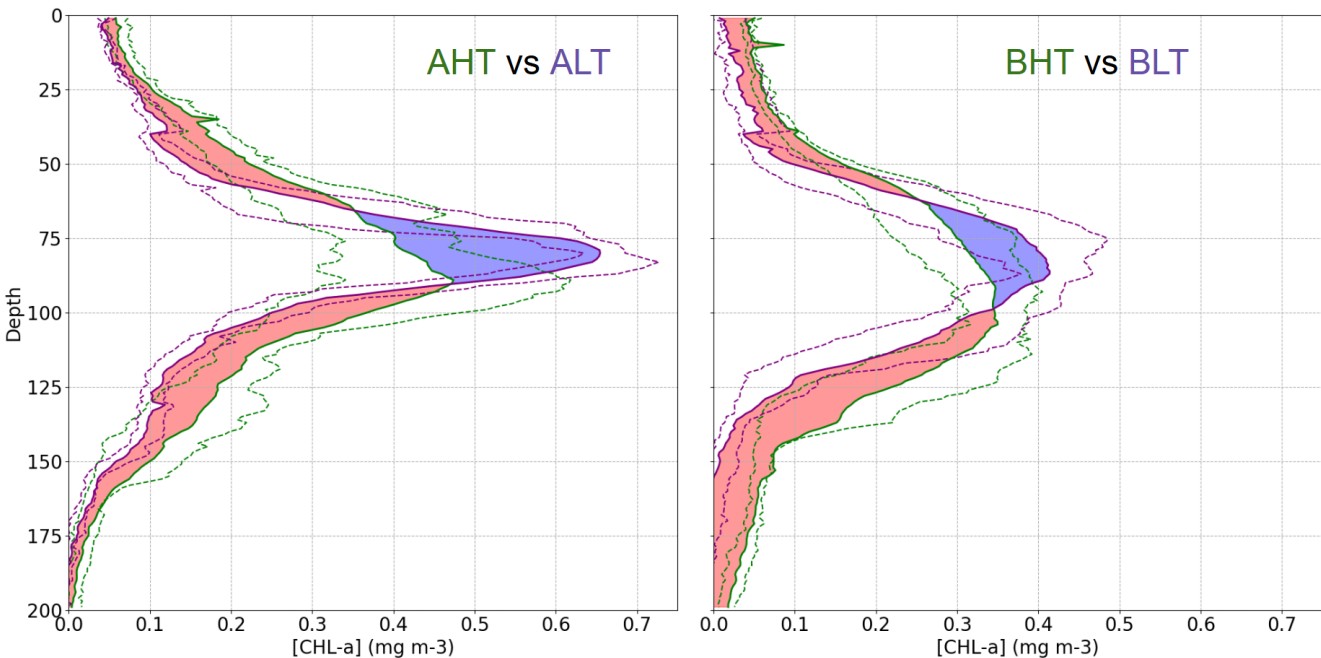


**Figure 10: Comparison of mean chlorophyll-a profiles during HT and LT periods. The purple dashed lines represent the interquartile range (IQR) for LT periods, while the green dashed area represents the IQR for HT periods. The red regions indicate where the mean chlorophyll-a concentration during HT exceeds that during LT, and the blue regions indicate the opposite (LT > HT).**



**Table 2: Summary statistics of chlorophyll-a of four key parameters: maximum chlorophyll-a concentration, chlorophyll-a peak thickness, mean chlorophyll-a content, and the depth of the deep chlorophyll-a maximum during HT and LT periods.**

| Period/features | Peak Thickness at 0.2 (mg m-3) | | | Depth of Max (m) | | |
|---|---|---|---|---|---|---|
| | Mean | Median | STD | Mean | Median | STD |
| AHT | 69.2 | 67.5 | 18.6 | 81.7 | 87 | 15.59 |
| ALT | 45.93 | 47 | 6.56 | 79.38 | 80.5 | 5.35 |
| BHT | 66.3 | 67 | 9.59 | 98.28 | 99.5 | 22.01 |
| BLT | 51 | 50.5 | 5.69 | 85.45 | 86.0 | 10.46 |
| Period/features | Max of chlorophyll-a (mg m-3) | | | Total Averaged Chlorophyll (mg -m3) | | |
| | Mean | Median | STD | Mean | Median | STD |
| AHT | 0.60 | 0.58 | 0.09 | 36.28 | 34.69 | 6.36 |
| ALT | 0.72 | 0.71 | 0.07 | 31.84 | 31.73 | 2.09 |
| BHT | 0.43 | 0.41 | 0.11 | 31.36 | 30.77 | 3.91 |
| BLT | 0.47 | 0.47 | 0.09 | 24.38 | 24.21 | 7.02 |





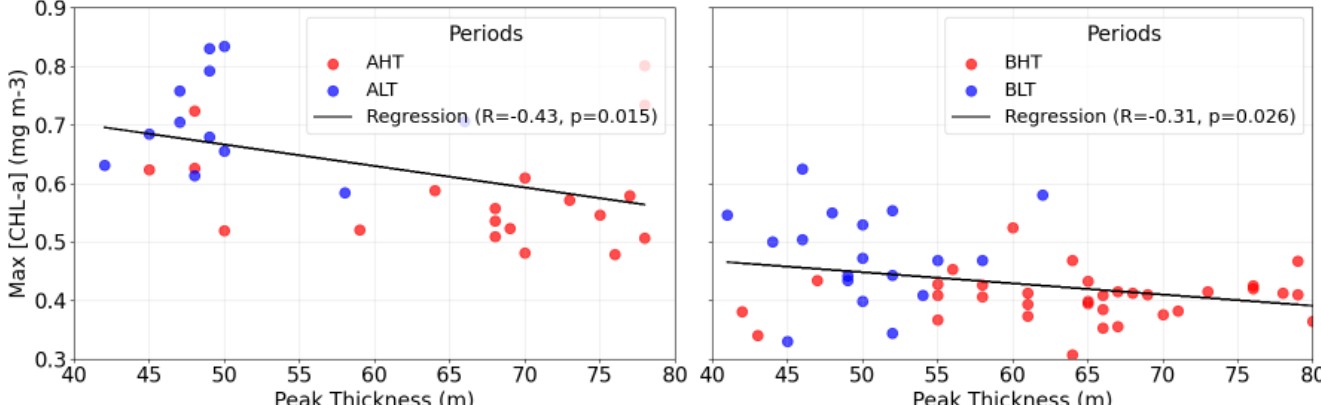

**Figure 11: Relationship between Chlorophyll peak thickness (m) and maximum chlorophyll-a concentration (mg m-3) during HT and LT periods.**

*Chlorophyll-a Diapycnal Redistribution*

The net chlorophyll-a loss observed in the Deep chlorophyll-a Maximum (DCM) layer between low tide (LT) and high tide (HT) was estimated at $\Delta CHL_{DCM} = -8.69$ mg m-2 or a 64% loss during period A ($-3.7$ mg m-2 or 21% loss during period B) as shown in table 3. This depletion was redistributed both upward and downward across isopycnal layers.

The downward turbulent flux reaching the deep isopycnal layer ($23.7 < \sigma_0 < 26.5$) was quantified as $\Delta CHL_{DEEP} = 5.14$ mg - m2 in period A (2.46 mg m-2 in period B). As this layer lay below the euphotic zone and did not support photosynthesis, biological consumption processes dominate ($\Delta SMS_{DEEP} < 0$) implying that the observed chlorophyll-a increase represented a minimum estimate of the turbulent flux to depth $\Delta CHL_{DEEP} <= \Delta Diff_{DEEP}$ (eq.7 ). Thus, turbulent fluxes from the DCM supplied approximately 57 % of the total chlorophyll-a increase observed in the deep layer. The turbulent flux toward the surface layer ($\sigma_0 < 23.53$) was consequently estimated as $\Delta CHL_{DCM} - \Delta Diff_{DEEP} = \Delta Diff_{SURF} = 3.55$ mg m-2 for period A ( 1.09 mg m-2 for period B). Thus, turbulent fluxes from the DCM supplied approximately 38% of the total chlorophyll-a increase observed in the surface layer. The total variation in surface chlorophyll-a content between LT and HT reached 7.99 mg m-2 in period A (8.07 mg m-2 in period B). After accounting for the turbulent input, the contribution of biological processes (production minus grazing) was calculated at $\Delta SMS_{surf} = 4.55$ mg m-2 for period A ( 6.98 mg m-2 for period B).



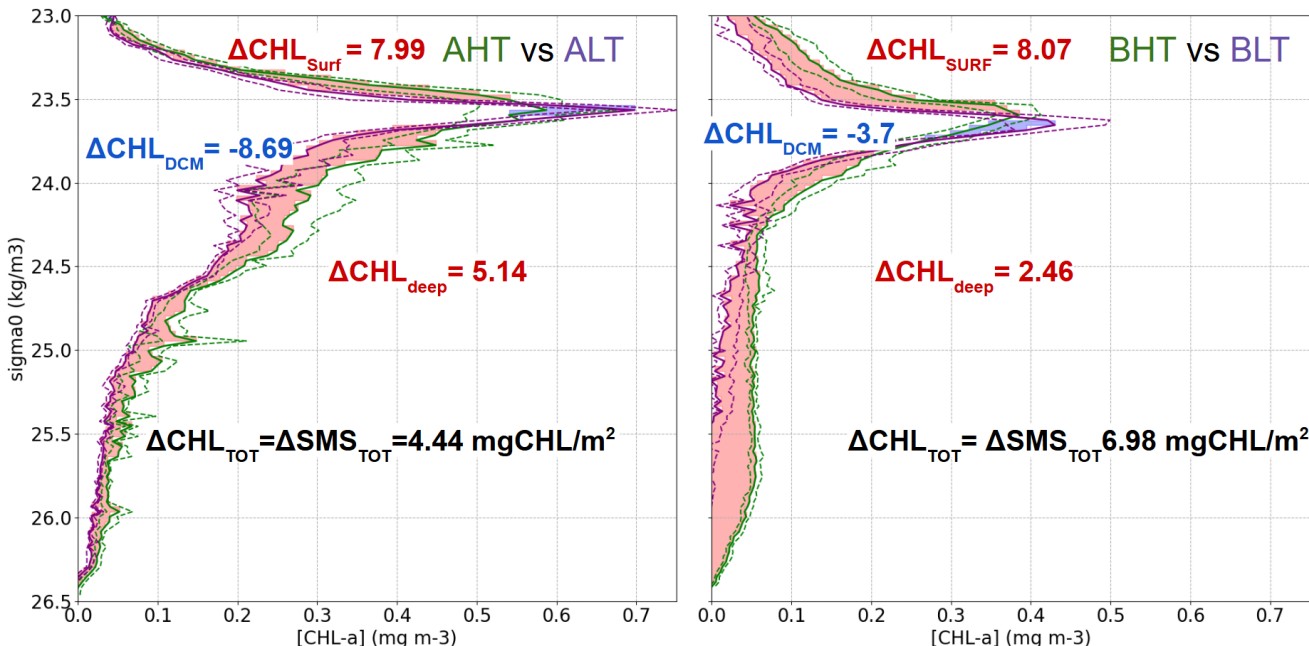

**Figure 12: Average vertical profiles of chlorophyll-a concentration (mg m-3) as a function of density (sigma0, kg/m³) for two regions: AHT vs ALT (left) and BHT vs BLT (right) based on density bins of 0.03. Shaded areas indicate differences in chlorophyll-a concentration between regimes, with red representing positive differences and blue indicating negative differences. Green and purple dashed areas represent the first and the third quartile for HT and LT periods.**

**Table 3 : Diapycnal statistics of integrated chlorophyll-a concentrations for each isopycnal layer**

| Layer | | A | | | | B | | |
|---|---|---|---|---|---|---|---|---|
| | Period | **Mean** | **Median** | **STD** | Period | **Mean** | **Median** | **STD** |
| **Surface - 23.53 (SURFACE)** | **AHT** | 17.37 | 15.92 | 4.23 | **BHT** | 10.96 | 12.01 | 4.06 |
| | **ALT** | 9.38 | 9.56 | 1.10 | **BLT** | 2.89 | 3.04 | 1.95 |
| **23.53 - 23.7** | **AHT** | 4.46 | 3.73 | 2.85 | **BHT** | 13.68 | 12.00 | 6.10 |



| (DCM) | ALT | 13.15 | 13.7 | 2.04 | BLT | 17.38 | 17.38 | 3.32 |
|---|---|---|---|---|---|---|---|---|
| 23.7 - 26.5 (DEEP) | AHT | 14.45 | 13.07 | 3.77 | BHT | 6.57 | 6.23 | 1.43 |
| | ALT | 9.31 | 9.15 | 2.25 | BLT | 4.11 | 3.66 | 2.54 |

## 4 Discussion

*AE1 as a mode water eddy*

In this study, AE1 was identified as a mode-water eddy with an isopycnal structure distinct from that of classical anticyclonic eddies. While classical anticyclonic eddies typically feature depressed isopycnals that form a bowl-like shape restricting nutrient access to the euphotic zone, cyclonic eddies are characterized by domed isopycnals that enhance nutrient uplift. Mode-water eddies, a particular type of anticyclone, combine both doming and depression of isopycnals and have been reported as productive systems in subtropical regions (Chelton et al., 2011; McGillicuddy et al., 2007). In the case of AE1, deeper, less dense isopycnals ($\sigma 0 \approx 23.5$) showed doming, while the upper, denser isopycnals ($\sigma 0 \approx 23.7$) displayed a bowl-like depression (Figure 6). Notably, during period B, AE1 was less productive compared to other periods, which contrasts with McGillicuddy et al. (2007) (Figure 9). This reduced productivity is likely due to AE1's greater depth; although uplift of isopycnals was observed, their vertical displacement remained insufficient to bring nutrients close to the euphotic layer. This finding suggests that the productivity of mode-water eddies is strongly influenced by their vertical positioning. Consequently, the deep chlorophyll maximum (DCM) and likely the nutricline were situated deeper, limiting light availability and constraining photosynthetic activity.

*Dominance of ITs Over Near Inertial Waves*

One may ask whether wind forcing contributes significantly to the observed variability. Figure 13 shows a spectral analysis of the glider data over the whole period of acquisition (one month). A clear and intense peak is observed at 1e5 close enough to the M2 12.25 hours (fig8 and fig 13). At the inertial period (approximately 7 days at 2°–4°N), a small peak at 5e3 is found suggesting that the wind-driven processes like near-inertial waves are less important in our region.

These findings support the conclusion that internal tides are the dominant oscillation that explain the intensified mixing that we estimate at the time scale of the month.





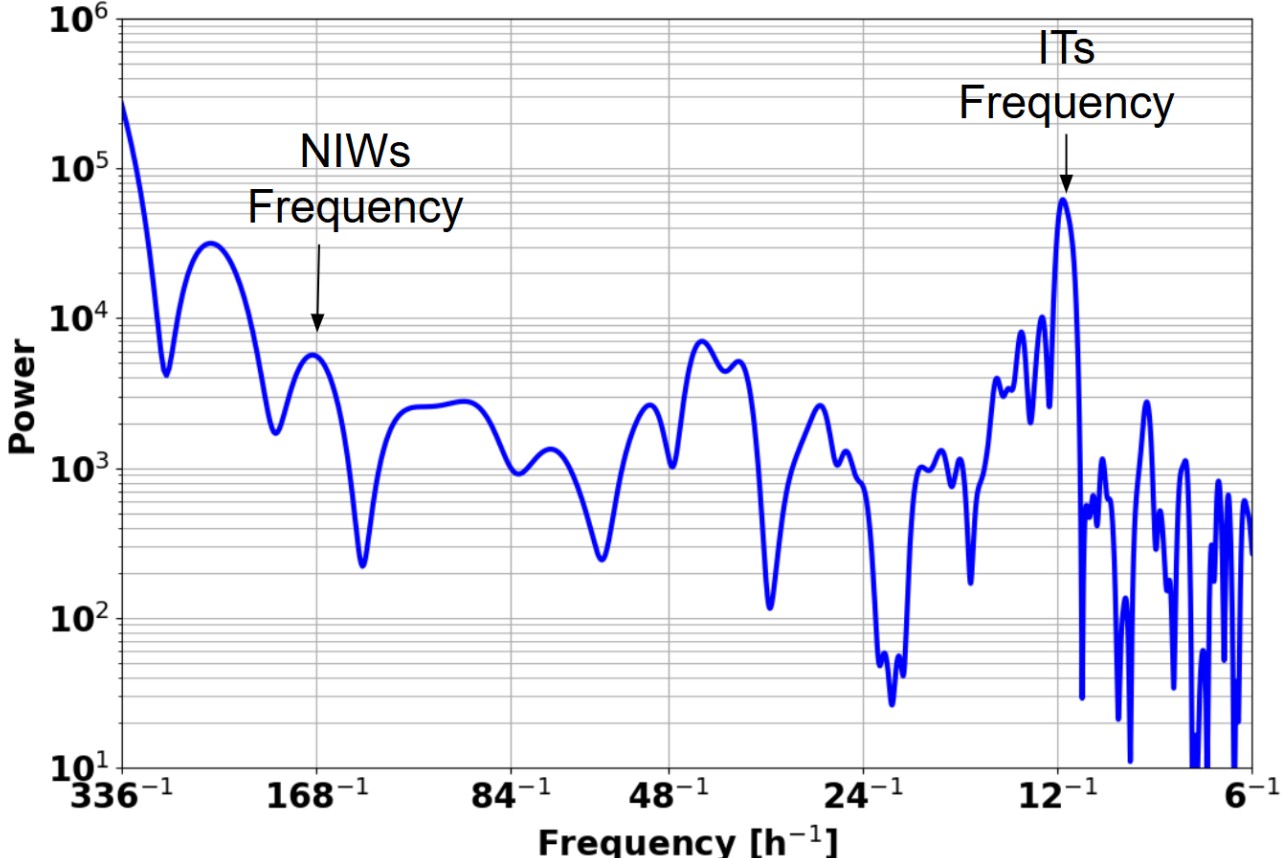


**Figure 13: Spectral Analysis of temperature from the whole time series.**

*Limits of Reversibility of LT and HT period and biological consequences*
A key limitation of our study is the limited number of sampling periods available to effectively assess the impact of ITs (ITs)
on the vertical distribution of chlorophyll. This constraint not only reduces statistical robustness but also restricts our ability
to generalize the observed patterns. An additional and important nuance lies in the specific sequence in which high tide (HT)
and low tide (LT) occur. In the available data for Period A, HT precedes LT (AHT/ALT), raising questions about the
reversibility of tidal effects. The biological impact of HT followed by LT may differ significantly from a reversed sequence
(ALT/AHT), particularly due to the lagged responses of phytoplankton communities.
This issue is especially relevant in our study region, which is characterized by regular and intense internal tide propagation,
generating near-continuous alternation between HT and LT phases. Rather than isolating the direct effect of individual ITs,
our approach focuses on a comparative analysis between HT and LT conditions, acknowledging the interconnectedness and

 

cumulative nature of these processes. Notably, the LT phase may still harbour biological communities that have benefited from
favourable mixing and nutrient supply conditions during the preceding HT phase. The extent to which this occurs depends on
the response time of the resident phytoplankton taxa.
In oligotrophic tropical waters such as ours, phytoplankton communities are typically dominated by small-sized cells, including
*Prochlorococcus*, *Synechococcus*, and various picoeukaryotes. These groups exhibit relatively rapid physiological responses
to environmental changes, particularly to nutrient enrichment. For example, *Prochlorococcus* can respond to pulses of nitrogen
or phosphorus within 12–36 hours (Moore et al., 2007; Partensky et al., 1999), while *Synechococcus* and picoeukaryotes tend
to show measurable increases in biomass within 24–72 hours (Calvo-Díaz et al., 2008; Fuchs et al., 2023; Scanlan et al., 2009;
Zubkov et al., 2000). In contrast, larger phytoplankton such as diatoms are less competitive under nutrient-poor conditions and
typically require more sustained or intense inputs to initiate growth, with response times ranging from 2 to 4 days (Falkowski
et al., 1998; Marañón et al., 2000). In our region, assuming that the biological response occurs within approximately one day,
this factor is unlikely to significantly influence our results. Consequently, the AHT/ALT sequence would yield similar
outcomes to an ALT/AHT sequence. The cumulative evidence from our findings (Fig. 9 and Fig. 11) supports this hypothesis,
suggesting that the phytoplankton species in our region exhibit a rapid response to light and nutrient availability. Further
validation of this hypothesis will be achieved through complementary analysis of AMAZOMIX cytometry data in future
studies.

*Chlorophyll turbulent vertical fluxes*

The diapycnal redistribution of chlorophyll-a observed between high tide (HT) and low tide (LT) phases (Fig. 12) is attributed
to mixing driven by internal tides (IT), under the assumption that large-scale background conditions remain similar between
HT and LT subperiods within the same overall period. Additionally, we argue that ITs dominate over near-inertial waves
(NIWs) in transporting biomass both upward into the euphotic zone and downward into deeper layers. This assumption holds
only if the water masses and active processes are comparable between the two phases. We verified that the water masses were
sufficiently similar to support this, but we acknowledge that this approach neglects possible contributions from other vertical
mixing processes, such as NIWs or frontal activity, which may differ between HT and LT. Longer time series would help
quantify these fluxes more precisely, beyond the limitations of a single-event analysis.
The deep chlorophyll maximum (DCM), located at the interface between nutrient-rich deep waters and the light-limited upper
layers (Ma et al(Ma et al., 2023)ma.), showed a net biomass loss between HT and LT, with 21% (period B) to over 60% (period
A) exported downward and the remainder redistributed upward. These findings partly align with the observations of Gaxiola-
Castro et al. (2002), who reported internal wave-driven upward transport of chlorophyll in the Gulf of California, increasing
surface biomass by around 40% during spring tides — consistent with the increases we observe. However, their study did not
quantify the downward flux, which in our case accounts for nearly half of the deep-layer biomass (~57%).





This redistribution has important implications for the trophic network. As Durham and Stocker (2012) have shown, thin
phytoplankton layers act as trophic hotspots, intensifying interactions among phytoplankton, zooplankton, and higher trophic
levels. The downward export of biomass not only contributes to the biological carbon pump but also reduces resource
availability for mesopelagic organisms. Meanwhile, the upward transfer enhances primary production, increasing surface
biomass by 14–29% and reinforcing upper trophic chains. Overall, these results highlight the crucial role of internal tides in
shaping marine trophic dynamics and underscore the importance of accounting for both upward and downward turbulent
fluxes.

**5 Conclusion**

This study provides new insights into the role of internal tides (ITs) in reshaping and homogenizing the vertical distribution of
chlorophyll-a and enhancing primary productivity off the Amazon shelf. Using a combination of satellite observations and
high-resolution glider data from the AMAZOMIX 2021 campaign, we show that ITs are key drivers of short-term vertical
chlorophyll variability.
The region is marked by dynamic interactions between major current systems, including the North Brazil Current (NBC), the
NBC retroflection, and the North Equatorial Countercurrent (NECC), as well as the presence of mesoscale and submesoscale
structures. During the glider deployment, five internal solitary wave (ISW) signals and a large anticyclonic eddy (AE1) were
detected by remote sensing. Combined with glider data, observations showed that AE1 locally reduced productivity by limiting
exchanges between surface and deep nutrient-rich layers.
Glider profiles revealed strong vertical isopycnal oscillations between 15 and 45 meters at semi-diurnal tidal frequencies. The
intensity of these oscillations allowed us to separate periods of strong internal tide activity (high tide, HT) from periods of
weaker activity (low tide, LT), which, under similar water mass conditions, provided a robust basis for comparing the effect
of internal tides on chlorophyll-a. Importantly, while ocean colour satellites are unable to resolve such fine-scale diurnal
variations, the glider was able to capture these dynamics, offering unique insights into the vertical redistribution of chlorophyll-
a.
Our results show that ITs redistribute chlorophyll-a vertically. This results in a thickening of the deep chlorophyll maximum
(DCM), increasing by 30–50% (~ +15 m) during high-tide periods, and a reduction in its peak chlorophyll-a concentration by
9–17% (~ –0.1 mg m-3). These effects are the results of both advection and mixing of the ITs.
First, the advection of the ITs induce vertical motion of the DCM, following the associated isopycnal displacement, which,
when averaged results in a larger DCM peak and combined with light conditions, may enhance primary production since the
light gradient is not linear with depth. Indeed, in the uplift condition, chlorophyll receives more light and increased Primary
production is expected; an uplift exposes the chlorophyll-a r light gain than the light loss caused by a downlift.





Second, the mixing plays a major role in reshaping the chlorophyll. Turbulent transport redistributes chlorophyll-a both upward
into the euphotic surface layers (accounting for ~40% of the chlorophyll content above the DCM) and downward into the
aphotic deep layers (about ~60% of the chlorophyll content below the DCM), with these fluxes originating from the DCM
pool and leading to losses of up to 65%.
In overall, the combined effect of advection and mixing, by improving both light availability and nutrient supply, leads to an
increase in the total chlorophyll-a content integrated over the whole water column by 14–29% during high internal tide phases
compared to low tide phases.
For future research, we recommend a more systematic use of gliders in oceanographic campaigns to enhance our understanding
of internal tides and their interactions with ocean biogeochemistry. We strongly advocate for the combined integration of
biological, physical, and turbulence sensors to better characterize the small-scale processes that control phytoplankton
dynamics and primary production.

**Data availability**

The AMAZOMIX glider data are available upon request by contacting the corresponding author.
Sentinel-1 SAR imagery: Copernicus Open Access Hub – https://scihub.copernicus.eu/dhus/
MODIS-TERRA/AQUA imagery: NASA Earthdata – https://earthdata.nasa.gov/
Chlorophyll-a and euphotic depth (Zeu): GlobColour via Copernicus Marine Service –
https://resources.marine.copernicus.eu/products
Absolute Dynamic Topography and geostrophic velocities: Copernicus Marine Service (SSALTO/DUACS)
https://data.marine.copernicus.eu/product/SEALEVEL_GLO_PHY_MDT_008_063/description
Bathymetry data: NOAA CoastWatch Data Portal – https://coastwatch.pfeg.noaa.gov/

**Authors contributions**


AKL: funding acquisition. AM and AKL, with assistance from ID,VP,ACS,AB,MA: conceptualization and
methodology. AM, with assistance from AB: data pre-processing. Formal analysis: AM with interactions from
all co-authors. Preparation of the manuscript: AM with contributions from all co-authors. This work is a
contribution to the LMI TAPIOCA (www.tapioca.ird.fr).



## Acknowledgements

M.A. thanks the support of the Brazilian research Network on Global Climate Change - Rede CLIMA (FINEP-CNPq 439 437167/2016-0)

The authors would like to thank the "Flotte Océanographique Française" and the officers and crew of the R/V Antea for their contribution to the success of the operations aboard the R/V ANTEA, as well as, all the scientists involved in data and water samples collection, for their valuable support during and after the AMAZOMIX cruise. We acknowledge the Brazilian authorities for authorising the survey,the National french parc of instrument (DT-INSU) for their instrument during the cruise and support in data analysis, as well as, the US-IMAGO from IRD for its help during the cruise and for biogeochemical data analysis.

## Financial support

This work is a part of the project "AMAZOMIX", funded multiple agencies : the "Flotte Océanographique Française" that funded the 40 days at sea of the R/V Antea, the Institut de Recherche pour le Développement (IRD), via among other the LMI TAPIOCA, the CNES, within the framework of the APR TOSCA MIAMAZ TOSCA project (PIs Ariane Koch-Larrouy, Vincent Vantrepotte, and Isabelle Dadou), the LEGOS and the program international Franco-Brazileiro GUYAMAZON (call No 005/2017). It is also part of the PhD Thesis of Amine M'hamdi, funded by the Fundação de Amparo a Ciência e Tecnologia do Estado de Pernambuco (FACEPE), under the co-advisement of Ariane Koch-Larrouy,Alex Costa da Silva, Isabelle Dadou and Vincent Vantrepotte.

## Competing interests

The authors declare that they have no conflict of interest.

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
