# Peer review of "Impact of Internal Tides on Chlorophyll-a Distribution and Primary 2 Production off the Amazon Shelf from Glider Measurements and 3 Satellite Observations"

_EGUsphere, 2025_

## Author Comment (AC1)

Dear reviewer,

We are very grateful for the constructive and relevant comments that allowed us improving this work.

Please find below our detailed responses to the comments.

General comments:

1. The results focus on a small area where it is known that internal tides have a significant influence on dynamics/mixing, termed a "hotspot" by the authors. I think something more needs to be said regarding the spatial distribution of such features; that is, whether the observations represent an extreme case or are typical. That might help clarify some of the implications of the work, e.g., whether internal tides are a key driver of variability as mentioned on line 39-41 in the abstract.

**Resp.: We agree with the reviewer that providing a broader spatial context is important to determine whether our observations represent an extreme case or a typical situation. The continental slope off the Amazon has been identified in previous studies (Baines, 1982; Magalhães et al., 2016; Tchilibou et al., 2022) as a major generation site for mode-1 internal tides. Our results indicate that internal tides, through vertical mixing, tend to homogenize the chlorophyll profile throughout the water column. This effect occurs only when two conditions are met: (1) the presence of active internal tides, and (2) a well-developed deep chlorophyll maximum (DCM). Similar processes have been documented by Gaxiola-Castro et al. (2002) in the Gulf of California and are also conceivable in other regions with strong internal tides propagation, such as the South China Sea and the Bay of Biscay. Finally, as our study area is not located directly at the generation site but rather in a region of strong internal tide propagation, our findings should be interpreted in this broader regional context . Lines 39 - 42 have been modified to make it clearer.**

2. I think there needs to some more discussion regarding the separation of spatial and temporal variations in the glider data. That is, inherently a glider that moves in space will capture variations in both space and time but without additional context it will be unclear which is more important. There are related elements already in the text; e.g. the discussion of eddy evolution and the location of the segments relative to the position of the eddy. But, I don't see any explicit mention of it. I think that is needed, even if the aliasing turns out to be minimal. Some discussion of how diurnal-weekly-monthly temporal variations, and the impacts on the observed spatial variability, would improve the manuscript.

**Resp.: We agree with the reviewer that separating spatial from temporal variability is a key challenge in glider datasets. To address this, we added a paragraph in the Discussion (L539–560) explicitly describing our approach: the transect was divided into**

**four hydrographically distinct regions (A–D) to isolate spatial variability, and within each region, HT and LT phases were compared under similar water mass properties. Averaging over multiple tidal cycles further reduced short-term variability. In our case, the 26-day glider record was averaged to daily resolution, which smooths out high-frequency variability (e.g., semi-diurnal and diurnal cycles) while allowing us to capture changes over 1–3 days associated with internal tide activity. The lower-frequency spring–neap modulation (~15 days) is only partially sampled within this time window, meaning that our analysis quantifies the short-timescale (1–3 day) component of the internal tide impact on chlorophyll redistribution rather than the full fortnightly cycle. This framework allows us to interpret the differences between HT and LT as representative of the mean internal tide signal, while acknowledging that some residual aliasing between spatial and temporal variability remains unavoidable**

3. I think some of the information provided in the methods section is not sufficient for the results to be reproducible. I have mentioned a few places where I think specific details are needed in the comments below.

   **Resp.: We have carefully addressed each of the specific points raised in the line-by-line comments and have added the requested clarifications to the Methods section.**

4. No direct turbulence or mixing estimates are used in this paper. While I do not think this is a problem, I think it should be clear earlier in the manuscript. Much of the language in the abstract/intro/early sections attributes changes to vertical mixing, which may be (likely is) the case but is not shown directly in the paper. I would recommend a careful edit of these sections so this is clear to the reader. Related to this (see my comment for L518), it is implied that vertical mixing is entirely a result of internal tides. While possible alternate contributing factors are clearly mentioned in the discussion, I think it would be helpful if it were mentioned earlier before presenting the results.

**Resp.: We have clarified in the Abstract (L33–35) that no direct turbulence measurements were collected in this study. We have also revised the wording in the early sections to indicate that the observed vertical redistribution of chlorophyll-a is consistent with tidally-driven cross-isopycnal exchanges, which represent the only physical mechanism to explain the transfer of biomass above and below the DCM and the observed variations.**

Line-by-line comments and suggestions:

L26-27 – I think this sentence is unnecessary. It is already stated later that they do modulate nutrient availability/productivity.

 **Resp.:  I've removed it**

L33 – "remarkable" compared to what? Please clarify

**Resp.: L.32 We agree with the reviewer that the term "remarkable" was subjective without a clear point of reference. We have revised the sentence to state explicitly that the 50% expansion refers to the difference between HT and LT states, removing the subjective qualifier.**

L36 – clarify what contributes to the other 62%

*Resp.: We updated it L.36 .At the surface, turbulent fluxes contributed 38% of the chlorophyll-a increase, while the remaining 62% resulted from net biological activity (primary production minus grazing). Both processes directly influence primary production.*

L55 – This sentence feels a bit disconnected; discuss how it influences climate variability, i.e. through air-sea interaction.

*Resp.: We have revised it to specify the mechanism, noting that surface-intensified mixing can alter sea surface temperature and thus modulate atmospheric convection and precipitation through air–sea interactions. l.63-65*

L96-97 – I think it would be helpful to add a sentence/references regarding the seasonality of internal tides and mesoscale features.

*Resp.: Done L.88 - 104*

L111 – clarify that this is in optimal conditions with no currents

*Resp.: Done*

L116 – change "thanks to" to "from" or "using"

*Resp.: Done*

L117 – strange wording. Reword "enabling to estimate"

*Resp.: Done*

L139 – change "imagery" to "images"?

*Resp.: Done*

L153 – extra space after 05

*Resp.: Done*

L157 – "merges"

*Resp.: Done*

L177 – the URL could be moved to a data statement at the end

*Resp.: Done*

L194 – What specific hydrographic properties were used to classify the data into these periods? Was this done objectively?

*Resp.: Updated L215-217The classification into hydrographic periods was based on distinct changes in water mass structure, identified from temperature, salinity, and potential density profiles. Transitions between periods were detected by examining vertical profiles and T–S diagrams for shifts in stratification patterns and salinity ranges across isopycnal layers. This classification was qualitative rather than based on an automated algorithm, relying on consistent, visually discernible features in the hydrographic data.*

Fig 1 – On a related note, there seem to be breaks between periods A&B and B&C. Are these transitional periods? Why were they not classified into any of the primary subregions?

*Resp.: Yes, these gaps correspond to transitional zones where the glider was moving between the hydrographically distinct regions defined in our analysis. Because the glider trajectory integrates both spatial and temporal variability, these transitional periods did not meet the criteria for homogeneous water mass properties used to define subregions A, B, C, and D. To ensure that comparisons between HT and LT were made under consistent hydrographic conditions, these transitional segments were excluded from the primary regional classification.*

L204 – How was the aggregating done? Is it assuming that every measurement within that depth range is treated the same? Or, was there some type of vertical averaging?

*Resp.: We have clarified in the Methods section (L224-238) that all measurements within the selected depth range (145–165 m) were treated equally, without applying vertical weighting. The individual measurements were concatenated into a single composite time series, resampled at 1 hour intervals, and linearly interpolated to produce a regular temporal grid before performing the spectral analysis. While no formal vertical averaging was performed, we assume that variability within this narrow depth band is coherent enough to be represented as a single aggregated signal.*

L210 – I'm curious how large of a contribution is expected from the SMS term? Is this a source of uncertainty?

*Resp.: The SMS term, encompassing biological sources and sinks (i.e., net primary production minus grazing), is indeed an important component in the chlorophyll-a balance. In our approach, we isolate the turbulent mixing contribution (Diff) using an isopycnal framework, which nullifies vertical advection. This method enables the*

*estimation of minimum turbulent fluxes and, by difference, the residual SMS term. For example, in Period A, biological processes account for approximately 57 % of the increase in surface chlorophyll-a, highlighting a substantial contribution from SMS. While we acknowledge that SMS estimation involves some uncertainty due to the lack of direct measurements of primary production and grazing, this residual approach is widely used for separating physical and biological contributions in observational studies. Importantly, even with this uncertainty, the relative magnitude of the SMS term consistently supports our interpretation that turbulent fluxes and biological processes jointly shape the observed chlorophyll-a variability*

L226 – How are high and low tidal forcing defined?

*Resp.: L263-265 Internal tides (ITs) are continuously present in the study region due to persistent barotropic tidal forcing over the topography. However, their intensity varies over time as a function of the spring–neap tidal cycle and local stratification conditions. In this context, High Tidal Forcing (HT) refers to periods within each observation window when internal tidal activity is most intense characterized by stronger isopycnal displacements. Conversely, Low Tidal Forcing (LT) corresponds to weaker activity with reduced vertical displacements. HT and LT are defined relatively to each other within each period.In short, HT corresponds to the period that is closer to spring tide conditions, while LT is the farther of the two.*

L233 – "integrated in DCM at the DCM" – I don't understand what this wording means. I think you mean integrated in depth within the layer.

*Resp.: Corrected L266_273*

Fig 2 – Nice schematic that shows how Chl changes vertically due to internal tides. I think the colors are somewhat ambiguous. It is unclear whether green refers to a) the sum of CHL and SMS or b) just Chl-a from SMS.

*Resp.: In Fig. 2, the green shading represents the potential impact range of SMS (biological sources and sinks), indicating where SMS could either increase or decrease chlorophyll-a concentrations. It does not represent the sum of chlorophyll-a and SMS, but rather the possible variation in chlorophyll-a attributable to SMS alone.*

L254 – Would a Spearman correlation analysis potentially be more appropriate, considering that I think we would not expect a linear relationship?

*Resp.: We agree with the reviewer that a Spearman ranked correlation is more appropriate in this case, as it does not assume linearity in the relationship. We have therefore recalculated the correlations using the Spearman method, and the revised values are now reported in the manuscript. updated L486*

L261 – missing space after the period

*Resp.: Done*

L268 – I disagree with this… it looks to me like euphotic depth Zeu decreases in the eddy core and increases on the eddy periphery, in a similar pattern to chlorophyll as described in the later text.

*Resp.: Ok Modified L309-310*

L277/279 – 11th / 12th

*Resp.: Done*

L277-280 – I'm not convinced there was "expansion" of the eddy. That seems, from the figure, to be an artifact of the cutoff ADT used to define the edge. Please reword to say that (or clarify if I am mistaken).

*Resp.: We acknowledge that the apparent "expansion" of AE1 is not solely related to an intrinsic growth of the eddy, but rather to a merging event with a neighbouring anticyclone (AE2) during this period. This type of process has been documented in previous studies (Thesis of Cori Pegliasco, 2017), where the progressive absorption of one eddy by another leads to an increase in the detected radius when using ADT-based contours. Since the dynamics of the merging are outside the scope of the present study, we did not develop this point in detail in this paper . Here is shown in white, while AE2 (not discussed in the main*

*text) is shown in blue.*

[Figure]

L288-300 – Following my previous comment, it looks like Zeu and chlorophyll are correlated within the eddy, but that this correlation seems to break down when outside the eddy. I think an explanation of this would be helpful.

**Resp.: Thank you for pointing this out. We have clarified in the manuscript that the estimation of Zeu follows the empirical relationship of Morel (1988) derived from surface chlorophyll-a concentrations cf L147, which explains the correlation observed between Zeu and chlorophyll-a within the eddy.**

L336 – Fig 6 appears to show that salinity was always above 35.5

Resp.: Yes, salinity values in Fig. 6 remain consistently above 35.5 psu throughout the study period. This indicates that the study site was not significantly influenced by freshwater from the Amazon plume during our observations, and thus plume-related stratification effects are negligible in this case.The threshold used to define euhaline waters comes from the Venice System for the Classification of Marine Waters (1958), which defines this category as having salinities between 30 and 40.

L333-352 – Nice summary. Much of the discussion on stratification is descriptive, however, and some of the trends mentioned in the text are not clearly apparent on the figure. For example, I do not clearly see more salinity stratification in region A than B, as is mentioned at L342. I think including quantitative information in a few places (i.e., dT/dz and dS/dz) would strengthen this section.

*Resp.: Thank you for the suggestion. The quantitative differences in stratification (dT/dz and dS/dz) are already reflected in the T–S diagram, which synthesizes the vertical gradients of temperature and salinity for each hydrographic period. This representation was chosen as it provides a compact view of both water mass structure and stratification differences between periods. However, we understand that some of the trends mentioned in the text are not fully apparent in Fig. 6. Following your suggestion (Next comment), we have moved the paragraph discussing the differences between the four periods earlier in the section, and clarified in the text the link between the T–S diagram and the corresponding vertical gradients*

L365-378 – The answers to some of my earlier comments are here. I think, perhaps, this should be moved earlier. I.e., discuss the four periods, then discuss their differences?

*Resp.: cf previous answer*

L368 – I think better to use 3 significant figures to be consistent here and in other places for the isopycnals

*Resp.: done*

L371 – "a distance was recorded" – odd wording; please rephrase

*Resp.: done*

Fig 8 – Add units on the y-axis (& for Fig 13). Also, it seems strange to me that the spectra are so smooth (or, perhaps I am mistaken). Was any smoothing done to the lines on this plot? If so, I would suggest to just plot the raw spectra.

*Resp.:Yes, the spectra in Fig. 8 were smoothed. Specifically, after resampling the time series to 1 h resolution, we applied a Hanning window to reduce spectral leakage, zero-padding to increase frequency resolution, and a 10-point moving average to smooth the resulting power spectra for clearer visualization. Following your suggestion, we have replaced these with the raw, unsmoothed spectra in the revised figure. here is raw spectra*

[Figure]

*up raw spectra for periods (AHT, ALT,BHT,BLT and D) / smoothed spectra (AHT, ALT,BHT,BLT and D)*

[Figure]

[Figure]

*up raw spectra (Full Time series 145_165m) / down smooth spectra (Full Time series 145_165m)*

L398-408 – It seems from the earlier plots that there is strong variability in surface chlorophyll. But this is not clear from Fig 9. Please explain this apparent discrepancy.

*Resp.: The apparent discrepancy arises from the difference in colorbar scaling. Surface chlorophyll values typically range between 0 and 0.1 mg/m³, whereas in Figure 9 the colorbar spans from 0 to 0.8. If the same scale used in Figure 9 were applied to the surface chlorophyll plots, the variations would appear minimal or even indistinguishable*

[Figure]

L419 – Odd wording. Maybe say "the peak is more pronounced"?

*Resp.: Done*

L422 – Was there significant temporal variability in the chl-a profiles? If so, perhaps a proportional criterion for thickness might work better?

*Resp.: We evaluated both a proportional criterion (width at half of the chl-a maximum) and a fixed threshold criterion of 0.2 mg Chl m⁻³ (see boxplots in Figure below). In both cases, the results consistently show a broader DCM during HT compared to LT, leading to the same interpretation: internal tides promote an increase in DCM thickness. We chose to retain the fixed 0.2 mg Chl m⁻³ criterion because it allows us to illustrate how internal tides enhance the vertical dispersion of elevated chlorophyll concentrations. For example, during LT conditions, chl-a concentrations above 0.2 mg m⁻³ are typically confined within a 45 m band, whereas under HT conditions they extend over ~69 m, indicating a substantial vertical redistribution.*

[Figure]

*Boxplot A (up) Variable Criterion / Boxplot B(down) 0.2 criterion*

L425 - While the relationship between thickness and high/low internal tide activity is very clear, I'm less convinced about the relationships between thickness and chlorophyll within the two IT regimes. It seems from Fig 11 that the high R values are because of peak thickness varies by much more than delta CHL, rather than a large change in chlorophyll. As suggested earlier, I think calculating a Spearman ranked correlation coefficient might be more appropriate, and would tell whether high values of Chla are associated with low values of thickness.

*Resp.: Thank you for the suggestion. We have recalculated the relationships between chlorophyll-a and layer thickness within each internal tide period using the Spearman ranked correlation coefficient. For period A, the correlation was negative and moderate (R = –0.44, p = 0.0125), indicating that higher chlorophyll-a values tend to be associated with*

*lower layer thickness. For period B, the correlation was weaker but still significant (R = –0.29, p = 0.0377). These results are now reported in the manuscript, providing a more robust assessment of the relationship without assuming linearity. Updated L471-472*

L428-430 – I think this is probably a stronger conclusion than the correlation coefficients (and is more clear in Fig 11). Maybe move earlier in the paragraph?

*Resp.: Ok done*

L447 – I'm a bit confused on how "chlorophyll-a loss" is calculated. From Fig 12, it does not look like the decrease in Chl at the peak is as high as 64%. Please clarify.

*Resp.: . The decrease in chlorophyll-a at the DCM may visually appear small in Fig. 12 because the profiles are plotted against density (σ), not depth. Since the DCM lies within the pycnocline, where density changes rapidly with depth, this region spans only a few meters vertically. As a result, even narrow shaded areas in σ-space can correspond to significant vertical gradients and chlorophyll losses. The relative loss (64%) is computed based on the difference between peak values in each condition, not the area under the curve, and reflects the sharp decrease at the DCM across a narrow vertical extent.*

L458 – Is there any SMS contribution to the DCM layer? I assume not based on the calculations in this paragraph.

*Resp.: Yes, biological processes such as grazing, photosynthesis do occur within the DCM layer. However, following the results of Ma et al. (2023), we assume that these processes are, on average, in dynamic equilibrium over the tidal timescales considered here. As a result, their net contribution (SMS term) within the DCM is assumed to be negligible in our calculations, and the observed changes are primarily attributed to physical processes such as tidally-driven cross-isopycnal mixing.*

L470-481 – nice summary of the differences between A and B.

*Resp.: thanks*

L493 – extra space between "ability"

*Resp.: done*

L518-524 –I think it would be useful to try to contextualize this more with existing literature – i.e. are there papers quantifying the impact of NIWs and fronts on chlorophyll. If not, are there any that have quantified physical turbulence parameters relating to these issues? I think having some additional background is important here, considering that the paper is based off of an implied assumption that the entirety of vertical mixing results from internal tides (which may be mostly true, but it would be helpful it this was put in context).

*Resp.: We would like to clarify that our study does not assume that all vertical mixing originates from internal tides. Rather, our methodology is designed to construct contrasted periods—high tide (HT) vs. low tide (LT)—so that the main varying physical driver is the internal tide. This allows us to estimate the delta in chlorophyll distribution attributable to more energetic tidal phases. We acknowledge that internal tides can coexist with other physical processes, such as submesoscale fronts or mesoscale eddies, and can even act synergistically with them. However, by comparing HT and LT conditions, we isolate the incremental effect of tides on vertical mixing and chlorophyll redistribution.*

*In many oceanic regions, near-inertial waves (NIWs) are a dominant source of mixing and nutrient supply, with documented bio-optical impacts (Granata et al., 1995; McNeil et al., 1999). In our study area, however, spectral analysis of the glider data shows a clear and intense peak at the M2 tidal frequency, while the inertial band (~7 days at 2°–4° N) displays only a weak signal. This is consistent with Kouogang et al. (2025), who reported that internal tides dominate vertical mixing over the Amazon shelf break year-round, with low near-inertial energy levels. Our results therefore quantify the tidal contribution to mixing in a background state where other sources of variability are minimal. While the role of submesoscale fronts in modulating mixing and primary production is well recognized, an assessment of their contribution in our study area is beyond the scope of this work.*

L575 – I'm not sure about the specific journal policy for this special issue regarding whether having data available upon request is acceptable.

*Resp.: We are currently in the process of depositing the dataset in Seanoe in accordance with the journal's data availability policy. The data will be made openly accessible upon submission of the revised manuscript.*

---

## Author Comment (AC2)

Dear reviewer,

We are very grateful for the constructive and relevant comments that allowed us improving this work.

Please find below our detailed responses to the comments.

Thank you for this opportunity to review this paper. This study investigates the impact of internal waves on a subsurface chlorophyll structure observed during a 26-day log glider deployment, complemented by satellite data. The manuscript presents a very interesting dataset and a compelling effort to explore the relationship between Chl-a concentrations and internal tides. However, several key elements in the methodology and interpretation of the results required further investigation and clarification. In particular:

1. Definition and identification of ISWs: I believe the introductions need more context and explanation of what internal solitary waves (ISWs) are and how they differ from internal tides. Mostly because ISW is a big part of the results and I believe there is some lack of clarity on how they are identified in the glider data. Do they have a different mixing diffusivity value compared to the tides? How do they relate to the separation of high tide vs Low tide analysis? In the results, the identification of ISWs—particularly in glider and satellite data—is unclear and inconsistent.

**Resp.: Following the suggestion, we have added a paragraph in the introduction (L48-60) clarifying the definition of Internal Solitary Waves (ISWs) and their relationship to Internal Tides (ITs). ISWs are nonlinear internal waves, shorter and more stable than ITs, which in our study region form primarily through the disintegration and dispersion of baroclinic internal tides. Unlike ITs, ISWs exhibit clear surface signatures, detectable in satellite imagery (MODIS sunglint or SAR), allowing us to identify their occurrence periods. In our study, ISWs are not directly included in the HT/LT analysis, which is based first on the semi-diurnal modulation of ITs identified from temperature spectra, and second on the classification into spring and neap tides. The ISW observations serve only as additional indicators of the presence and propagation of ITs in the region, as well as of the dominant propagation modes. Regarding their role in vertical mixing, ISWs, owing to their stability and ability to propagate over long distances, are generally less dissipative than ITs near their generation site. They can, however, contribute locally to mixing when they break, but this contribution was not quantified in our study.**

2. Assumptions about mixing and chlorophyll: A central conclusion of the paper is that differences in chlorophyll concentrations between high tide and low tide are due to physical mixing, but this assumption is not entirely justified in the methods and excludes potential biological processes within the DMC. I think the paper still has good results, but without turbulence or mixing data, the inferred mechanisms require stronger connection to prior work or clearer acknowledgment of uncertainty.

**Resp.: We acknowledge the need to better justify the assumption that differences in chlorophyll between high-tide (HT) and low-tide (LT) conditions are due to physical mixing. The limitations inherent to an observation-based approach compel us to make explicit assumptions in the interpretation of our results. Our reasoning follows the conceptual framework described by Ma et al. (2023) and earlier studies in the South China Sea (B. Chen et al., 2013) and equatorial Pacific (Landry et al., 2011), in which the Deep Chlorophyll Maximum Layer (DCML) represents a transition zone where light and nutrients jointly limit photosynthetic rates, and where phytoplankton growth and loss rates are in dynamic equilibrium. Within such an equilibrium state, the only physical mechanism capable of modifying chlorophyll concentrations within an isopycnal layer is turbulent mixing. While direct turbulence measurements are not available in our dataset, our analysis quantifies the contribution of this mixing to the observed differences in chlorophyll between HT and LT.**

3. Glider data processing and resolution: The methods section lacks detail on how glider data were interpolated, gridded, or treated before spectral analysis. Details about dive depth, vertical resolution, and time-series construction are critical to evaluating the strength of the results. This is particularly important for the spectral analysis

**Resp.: All scientific and navigation data were linearly interpolated to 1-second intervals to align science variables with the glider's main processor clock. This step introduces minimal additional noise, as the vertical displacement of the glider over one second is typically < 0.2 m. In the vertical, data from each dive profile (yo) were binned and averaged into 1 dbar intervals, and then linearly interpolated to produce uniformly gridded vertical profiles. These gridded profiles were used in all subsequent analyses, including stratification diagnostics and vertical chlorophyll characterization. After the standard GEOMAR Toolbox gridding procedure (1 dbar vertical binning and timestamp alignment), we applied a second linear interpolation in time to project the variables onto a regular temporal grid. This interpolation was performed independently at each depth level using valid (non-NaN) observations, ensuring a complete and consistent depth–time matrix for variability analyses. Importantly, spectral analyses were performed using the original gridded data prior to this second temporal interpolation to avoid any potential alteration of the spectral signal.**

4.  Justification of assumptions and definitions: Further justification and clarification of how key periods, depths, structures, and thresholds are defined throughout the study is needed to strengthen the interpretation of the results.

**Resp.: We thank the reviewer for highlighting the need for clearer justification of the definitions used for periods, depths, structures, and thresholds. We have revised the Methods section to explicitly detail how hydrographic periods were identified (based on consistent T, S, and $\sigma_0$ structures and visually discernible transitions in T–S diagrams), how transitional zones were treated (excluded from primary comparisons to ensure water-mass homogeneity), and how high/low tidal forcing phases were defined relative to local spring–neap variability and isopycnal displacement amplitude. We have also clarified the rationale for the selected depth ranges and vertical thresholds (e.g., 0.2 mg m$^{-3}$ criterion for DCM thickness) and indicated where alternative definitions were tested and yielded consistent results (e.g., proportional vs. fixed threshold criteria). These methodological clarifications, combined with the changes described in our line-by-line responses, ensure reproducibility and transparency while maintaining focus on the main scope of the paper**

Overall, I think this work has great potential to contribute to the literature of the region, but it needs major revisions to improve its readability and impact of its results. Below I describe in detail major comments and minor comments:

Major Comments:

Lines 47-52: The introduction of the ISW theory might need some work. The acronym is used before explaining what it is, and these sentences appear out of order. SWs are mentioned frequently throughout the paper, so it would be helpful to include more background here—how they are generated and how they differ from internal tide

**Resp.: In the revised manuscript (L48 - 60), we have reorganized the introduction to define internal solitary waves (ISWs) before using the acronym and to clearly distinguish them from internal tides (ITs). We now provide additional background on ISW generation mechanisms, including their formation from the nonlinear transformation of ITs as well.**

Lines 115-124: Throughout the study, there were different ways of using the glider data (surface comparison with the satellite, spectral analysis etc), which I think is excellent, but it's not clear from the methods how the data were interpolated (if it was) or gridded. Also, what was the maximum dive depth? Later, it's mentioned the glider does 12 profiles per day, with 2 hours per profile (Line 203), suggesting it's not reaching 1000 m. More detail on glider operations would help readers understand the interpretation of the data analysis

**Resp.: Done L117-140 however The glider performed dives reaching a maximum depth of ~950 m, completing on average 12 profiles per day (~2 h per profile).**

Line 233: The assumption that differences in chlorophyll a between high and low tide are due solely to mixing needs more support. What are the limitations of this assumption? Does this imply ΔSMS_dcm = 0? Since turbulent mixing was not measured, it would strengthen the argument to connect with prior work from the region that documented internal wave-driven mixing or estimated diffusivities consistent with your interpretation. Including possible mechanism (shear-driven turbulence? )

**Resp.: cf the comment 2 For the case $\Delta SMS_{dcm} = 0$ (see Comment 2).Although no direct turbulent mixing measurements were collected in our study, our interpretation is consistent with recent observations from the region. Kouogang et al. (2025) documented, in the area corresponding to our Region A, that internal tides (ITs) dominate vertical mixing off the Amazon shelf break, with dissipation rates reaching $10^{-6}$ W/ kg near IT generation sites and still substantial values (~$10^{-8}$ W/ kg) hundreds of kilometers offshore. Microstructure analyses revealed that IT shear contributed up to 60 % of total shear-driven turbulence, and that elevated dissipation in the far field was often associated with large-amplitude internal solitary waves (ISWs) generated by constructive interference of IT rays. These results support our interpretation that the vertical chlorophyll redistribution we observe in Region A is primarily driven by IT-induced shear-driven turbulence, with ISWs playing a secondary but locally significant role. We have now included a reference to Kouogang et al. (2025) in Lines 270–272 to support our interpretation.**

Figure 2: The diagram is hard to interpret. There is no context for why CHL_LT shows a larger peak than CHL_HT. After reading the results, this becomes clearer, but at this point is hard to follow the logic. Why are there two green lines?

**Resp.: *In Fig. 2, the green shading represents the potential impact range of SMS (biological sources and sinks), indicating where SMS could either increase or decrease chlorophyll-a concentrations. It does not represent the sum of chlorophyll-a and SMS, but rather the possible variation in chlorophyll-a attributable to SMS alone.We have updated the legend to clarify this point and facilitate understanding.***

Line 279-280: Is this growth of the eddy observed here typical this region? The speed in which it grows appear fast, but I am not be familiar with eddy activity here.

**Resp.: *We acknowledge that the apparent "expansion" of AE1 is not solely related to an intrinsic growth of the eddy, but rather to a merging event with a neighbouring anticyclone (AE2) during this period. This type of process has been documented in previous studies (Thesis of Cori Pegliasco, 2017), where the progressive absorption of one eddy by another leads to an increase in the detected radius when using ADT-based contours. Since the dynamics of the merging are outside the scope of the present study, we did not develop this***

*point in detail in this paper . Here is shown in white, while AE2 (not discussed in the main text) is shown in blue.*

[Figure]

Line 315-317: The identification of ISWs in Figure 5d is unclear. Are these timestamps of whent hey are observed in the glider or satellite data? If satellite, how is timing assigned? ? There also seem to be solitons near the spring-neap transition, which complicates the assertion that ISWs align with spring tides. This relationship and its time scale need further clarification—maybe add more context in the introduction.

**Resp.: In the revised manuscript, we clarify that the ISW occurrences marked in Figure 5d correspond to the exact timestamps of their detection from satellite imagery (SAR or sunglint MODIS), with acquisition times provided in the satellite data products. We have expanded the explanation of their relationship with the spring–neap cycle: in this region, ISWs are generated primarily by the nonlinear steepening of internal tides forced by barotropic tidal flow, which are typically stronger during spring tides. Nevertheless, ISWs can also occur during neap tides or transitional phases, albeit less frequently, which explains the detections near spring–neap transitions. This additional context has been incorporated into the introduction(cf commentary 1) to better describe the physical link and time scales involved. Furthermore, the term *crests* has been replaced with *wave packet detected* in the table 1 to avoid misunderstanding**

Table 1: Figure 5 seems to show two crests on September 9—was a height threshold used to identify crests?

**Resp.: No amplitude or height threshold was applied for crest identification. In Table 1, the "crest" column does not refer to individual wave amplitude but if we succeed to identifie internal solitary wave train. For example, on 9 September, the satellite imagery revealed a single ISW packet. We have clarified this wording in the table caption and main text to avoid confusion.**

Line 370: The phrase "well-defined T/S stratification" needs clarification. Do you mean stronger or weaker stratification? Is it more linear? Or does it refer to T and S both increasing or decreasing with depth?

**Resp.: This sentence has been removed due to its ambiguity. It referred to the presence of a lens-like feature, visible in Figure 6, with homogeneous TTT, SSS, and $\sigma_0$. Moreover, the sections *Transect Divided into Four Periods* and *Near-Surface Hydrography* were reorganized at the request of Reviewer 1, and we preferred to avoid redundancy(cf L393).**

Line 371-373: I think Period C seems to be fresher than B within the 24 –24.8 mass?

**Resp.: We agree with the reviewer's observation. Period C is indeed fresher than Period B within the 24–24.8 $\sigma_0$ layer. We have corrected the text accordingly to reflect this difference.**

Line 386: There seems to be an assumption that ISWs coincide with tidal peaks—but this is not apparent in Figures 5 or 6. For instance, an ISW is labeled on 13 Sept, but no large oscillation is visible. Also, which peaks are being referenced? (See earlier comment about identifying ISWs.)

**Resp.: We agree with the reviewer that not all identified ISWs in Figure 5d coincide with visible large-amplitude isopycnal oscillations in Figures 5 or 6. This is the majority of detections (5 out of 6) occurred during spring tide phase (yellow), which is consistent with the known stronger generation of internal tides during these phases. In this context, "peaks" refers to isopycnal crests associated with internal tide-induced vertical displacements, which in turn can steepen into ISWs.**

Line 386–387: The drop in surface temperature during spring tides (sections A and C) could be due to other causes—e.g., position relative to NECC or eddy edges—rather than tides alone. This sentence seems to imply that the tides drive this drop in temperature, but is this through mixing? Or another process?

**Resp.: The associated drops in temperature are consistent with previous studies off the Amazon shelf showing cooling above the thermocline and warming below during IT activity (Assene et al., 2024). We have revised the text to clarify these points and avoid overgeneralizing the ISW–spring tide relationship**

Line 388: How was the glider data used and prepared to create these FFT? Were they interpolated to a uniform time series?

The sentence "A Fast Fourier Transform (FFT) analysis of isotherms (145–165 m) confirms the semi-diurnal modulation of these oscillations" is unclear in its current form and would benefit from further clarification.From the results, I'm inferring that the FFT is examining the variability of vertical displacement of an isotherm, not the variability of temperature at a fixed depth. Is this correct? Maybe adding units to the spectrum figure will also help clarify this.  was some form of averaging or stacking performed across this depth interval that makes the plot so smooth?  How was the glider data prepared for the FFT? Was it interpolated to a specific depth? Was it bin averaged? How would this impact your results? A more precise description of the methodology—especially the variable being spectrally analyzed and how it was derived—would greatly improve the reader's ability to interpret the results and evaluate the evidence for semi-diurnal modulation.

**Resp.: In the revised manuscript L225 - 238, we have clarified the methodology used to produce the FFT. The analysis was indeed based on temperature variability at a fixed depth range (145–165 m), which was chosen because it corresponds to the layer of largest isopycnal vertical displacement. All measurements within this range were concatenated into a composite 1D time series, assuming coherent variability within the layer. The glider data, initially sampled at irregular intervals due to profiling motion, were resampled to a regular 1 hour grid using averaging followed by linear interpolation. This ensured temporal uniformity for the FFT while preserving sub-tidal variability. The time series was then detrended to remove long-term trends, and the FFT was applied to identify dominant oscillation frequencies. We have also clarified in the text that the FFT examines variability of temperature in this depth range (as a proxy for isopycnal displacement) rather than the vertical displacement of a single isotherm. Units have been added to the spectral density figure to improve interpretability.**

Line 448: Could changes in DCM chlorophyll be due to biological responses, not just physical mixing? This relates to the concern above about the mixing assumption. The equation on line 454 is also unclear and needs more explanation

**Resp.:  We agree that the presentation of the equation on line 499 could be clearer. The underlying idea is straightforward: the loss of chlorophyll-a in the DCM layer during HT relative to LT is assumed to be entirely due to turbulent fluxes. By conservation of mass, this turbulent loss from the DCM is redistributed upward to the surface layer and downward to the deep layer.**

In our formulation (Eqs. 5–7), $\Delta CHL_{SURF}$ corresponds to the turbulent gain in the surface layer plus any biological contribution, and the turbulent comp5-7). Thus, the term on line 454, $\Delta CHL_{DCM} - \Delta Diff_{DEEP}$, simply represents the turbulent flux from the DCM that is directed upward into the surface layer. We have revised L497 the text to explicitly link this equation back to Eq. 5-7 and to clarify that it follows directly from the mass-conservation assumption applied to the turbulent redistribution between layers.

Line 474 -475: The phrasing is confusing: "deeper, less dense" or "upper, denser"? Clarify what part of the eddy is being described. Additionally, the reference to McGillicuddy et al. (line 478) requires more context. Greater depth compared to what?

Resp. : We have revised the sentence for clarity L519-521. In McGillicuddy's framework, the doming part of an anticyclonic system can drive isopycnal uplift, potentially enhancing biological productivity by injecting nutrient-rich waters into the euphotic zone. In our case, while such isopycnal uplift is indeed observed within AE1, the anticyclone core appears too deep for this mechanism to significantly increase productivity, likely because the uplifted layers remain below the light-limited depth.

Lines 482-488: I think these results should be added in the section of the results where the authors do the spectrum analysis. Its presence here is unexpected and underdeveloped. Maybe other questions can be answered from these distinctions: why is it important to distinguish between these two types of oscillations (wind forcing, length scales, etc)? Have other papers discussed these differences, and do the results agree with your findings? Also, how was the spectrum in Figure 13 produced? The same as Fig. 8 but longer time series? What is the error bar, How many spectra were averaged, and what are the error estimates?

Is there any filtering applied to the data? Are they the same depth as Fig 8?

Resp.: We thank the reviewer for these suggestions. The main focus of this paper is on the impact of internal tides on chlorophyll-a, and a full comparison with other types of oscillations (e.g., near-inertial waves) would be beyond the scope of the present study. However, we anticipated that some readers might question the potential role of wind-forced near-inertial waves in driving mixing. For this reason, we retained Figure 13 in the discussion, not as a primary result, but as supporting evidence to address this possible question from the readership. The spectrum shows that the dominant oscillations are at the M2 tidal frequency, with only a minor peak at the local inertial frequency, consistent with Kouogang et al. (2025) for the Amazon shelf break.

We have now clarified in the manuscript that the spectra in both Figures 8 and 13 were produced using the same methodology: temperature data between 145–165 m, resampled at 30 min intervals, detrended, and processed using a Fast Fourier Transform with a Hanning window applied to reduce spectral leakage. Figure 13 uses the full one-month glider time series, whereas Figure 8 uses shorter sub-periods

**corresponding to the defined study periods. No additional filtering was performed. Only one spectrum was computed from each aggregated time series, so no error bars are provided. While the analysis of Figure 13 has now been moved to the spectral results section to improve logical flow, its inclusion is maintained to explicitly address potential alternative explanations for vertical mixing raised by readers.**

Line 504-416:This ecological context is appreciated and very usefull—perhaps connect it more explicitly to the tidal vs. near-inertial forcing context in terms of timescale?

**Resp.: We agree that linking the ecological context to the relevant physical forcing timescales would be valuable. Near-inertial pumping occurs when spatial and temporal variations in wind forcing generate inertial oscillations, leading to alternating divergence and convergence zones in the mixed layer that drive vertical displacements of its base (Gill, 1984). This process can supply nutrients to the euphotic zone on inertial timescales (~7 days at our latitude), which differ from the semi-diurnal timescales of internal tides. However, spectral analysis of our glider records indicates negligible energy in the inertial band compared to the strong M2 tidal peak, suggesting that near-inertial processes were not a significant driver of vertical mixing during our observations. This is consistent with recent findings by Kouogang (2025), who showed that internal tides dominate vertical mixing over the Amazon shelf break, with near-inertial energy levels remaining low throughout the year. Therefore, while near-inertial pumping is an important process in other oceanic regions, its detailed investigation lies beyond the scope of the present study.**

Minor comments

Line 97: Add more information about why Sep and Oct 2021 were an optimal period for IT activity

**Resp.:  Added L88-96**

Line 214: A closing parenthesis is missing from the equation.

**Resp.: Done**

Line 217-219: If previous studies used a similar derivation for these equations, please cite them.

**Resp.:Thanks for the comment we integrated in the paper L240-242.The set of equations used in this study is an adaptation of the NPZ-type framework presented in Franks (2002) to our observational case, in which vertical turbulent fluxes are primarily driven by internal tides. In this adaptation, the vertical mixing term is explicitly linked**

**to the cross-isopycnal turbulent diffusivity estimated for HT and LT phases, and the three-layer structure (surface, DCM, deep) is defined according to our in situ density and chlorophyll profiles. To our knowledge, this specific formulation has not been used in previous IT-focused studies.**

Line 233: The terminology DCM (ΔDiff_DCM) is not clear to me, i suggest explaining what Diff(DCM) means.

**Resp.: Here, $\Delta\text{Diff}_{DCM}$ refers to the change in depth-integrated chlorophyll-a (mg m$^{-2}$) within the DCM layer attributable to turbulent diffusive fluxes. We now precise it L.267-269**

Line 267: Add reference to figure 3e-h

**Resp.:Done L307**

Line 421: Why was 0.2 mg/m³ used as the threshold for chlorophyll peak thickness?

**Resp.: Because the minimum of value of DCM was around 4 and 0.2 is the half of for**

Suggestions and minor questions

Line 79: Consider referencing Figure 1 to help the reader visualize the study area.

**Resp.: done**

Line 336: Is there a specific reason why you use 35.5 as the euhaline threshold? As someone unfamiliar with this region, this seems a high threshold.

**Resp.:The threshold used to define euhaline waters comes from the Venice System for the Classification of Marine Waters (1958), which defines this category as having salinities between 30 and 40.**

Line 338: I suggest referencing the black lines in Figure 3a when describing the cross of AE1

**Resp.: Thank you for the suggestion. We have updated the text to explicitly reference the black lines in Figure 3a when describing the transect across AE1.**